# ATAD3A oligomerization promotes neuropathology and cognitive deficits in Alzheimer's disease models

Yuanyuan Zhao [1,5], Di Hu [1,5], Rihua Wang[1,5], Xiaoyan Sun[1], Philip Ropelewski[1], Zita Hubler[2], Kathleen Lundberg[3], Quanqiu Wang[4], Drew J. Adams[2], Rong Xu [4] & Xin Qi [1✉]

Predisposition to Alzheimer's disease (AD) may arise from lipid metabolism perturbation, however, the underlying mechanism remains elusive. Here, we identify ATPase family AAA-domain containing protein 3A (ATAD3A), a mitochondrial AAA-ATPase, as a molecular switch that links cholesterol metabolism impairment to AD phenotypes. In neuronal models of AD, the 5XFAD mouse model and post-mortem AD brains, ATAD3A is oligomerized and accumulated at the mitochondria-associated ER membranes (MAMs), where it induces cholesterol accumulation by inhibiting gene expression of CYP46A1, an enzyme governing brain cholesterol clearance. ATAD3A and CYP46A1 cooperate to promote APP processing and synaptic loss. Suppressing ATAD3A oligomerization by heterozygous ATAD3A knockout or pharmacological inhibition with DA1 restores neuronal CYP46A1 levels, normalizes brain cholesterol turnover and MAM integrity, suppresses APP processing and synaptic loss, and consequently reduces AD neuropathology and cognitive deficits in AD transgenic mice. These findings reveal a role for ATAD3A oligomerization in AD pathogenesis and suggest ATAD3A as a potential therapeutic target for AD.

[1] Department of Physiology & Biophysics, Case Western Reserve University School of Medicine, Cleveland, OH 44106, USA. [2] Department of Genetics, Case Western Reserve University School of Medicine, Cleveland, OH 44106, USA. [3] Proteomics Center, Case Western Reserve University School of Medicine, Cleveland, OH 44106, USA. [4] Center for Artificial Intelligence in Drug Discovery, Case Western Reserve University School of Medicine, Cleveland, OH 44106, USA. [5] These authors contributed equally: Yuanyuan Zhao, Di Hu, Rihua Wang. ✉email: xxq38@case.edu

Alzheimer's disease (AD) is the most common age-dependent neurodegenerative disease with unknown etiology. AD is characterized by the accumulation of amyloid deposition, neurofibrillary tangles, synaptic loss, and progressive cognitive decline[1]. Since the discovery of AD over 100 years ago, the underlying mechanisms of cellular damage and cognitive deficits remain elusive. As a result, current AD therapies are poorly effective and limited to acetylcholine or N-methyl-D-aspartate glutamatergic mechanisms that provide only mild symptomatic benefit and fail to slow disease progression[2]. Thus, the identification of novel therapeutic targets is imperative for developing disease-modifying therapies.

Mitochondria and endoplasmic reticulum (ER) are highly interconnected. They physically interact to form specific microdomains called mitochondria-associated ER membranes (MAMs), where the outer mitochondrial membrane is close to the ER (i.e., within 10–30 nm)[3]. The MAMs are involved in many key metabolic functions[4], including cholesterol metabolism[5], lipid synthesis and trafficking[6], mitochondrial dynamics[7], calcium homeostasis[8], and autophagy[9]. All these functions are altered in neurodegenerative diseases, including AD. Indeed, the integrity of the MAMs is significantly impaired in AD animal models and patients, manifesting as a hyperconnectivity of the MAMs[8,10]. MAM-resident proteins inositol 1,4,5-trisphosphate receptor (IP3R) and long-chain acyl-CoA synthetase (FACL4) increase in various AD experimental models and the postmortem brains of AD patients[11,12]. Polymorphisms in mitofusin 2 and sigma non-opioid intracellular 1-receptor 1 (SigmaR1), two MAM proteins, correlate with the risk of developing AD[13,14]. Moreover, the amyloid precursor protein (APP) processing γ-secretases, presenilin-1 and presenilin-2, are highly enriched in the MAMs relative to other cell compartments, such as the plasma membrane, mitochondria, and ER[10]. These findings highlight the role of MAMs in amyloidogenesis. In addition, the ε4 allele of apolipoprotein E, the most common genetic risk factor of late-onset AD, upregulates MAM activity[15]. Thus, perturbed MAMs are a key event in AD pathogenesis and may serve as a common convergent neurodegenerative mechanism[16]. However, the factors that induce MAM hyperconnectivity in AD are poorly understood, and whether manipulation of impaired MAMs affects AD progression has not been explored.

Perturbations in lipid homeostasis are another feature of AD. Accumulation of cholesterol has been observed in senile plaques and affected brain areas of AD patients[17], and is associated with region-specific loss of synapses[18]. A growing number of animal studies have consistently demonstrated that hypercholesterolemia leads to dysfunction of the cholinergic system, cognitive deficits, and amyloid and tau pathology[19,20], all of which strongly support a role for cholesterol disturbance in AD. In familial and sporadic AD subjects, increased cholesterol esters can be detected in the lipid raft-like MAMs[10]. Hyperactivity of MAM tethering causes cholesterol accumulation and synaptic loss and is associated with cognitive deficits[21]. In addition, the cleaved product of APP (i.e., C99) accumulates at MAMs, where it impairs mitochondrial bioenergetics, disrupts cellular lipid homeostasis, and causes alterations in membrane lipid composition commonly observed during AD pathogenesis[22,23]. Despite these findings, the mechanism that links MAM impairment, cholesterol accumulation, and amyloidogenesis in AD remains elusive.

ATPase family AAA-domain containing protein 3A (ATAD3A) is a nuclear-encoded mitochondrial membrane protein that belongs to the AAA$^+$-ATPase protein family. ATAD3A has a unique structure with a C-terminus that includes a conserved ATPase and is located in the mitochondrial matrix and an N-terminus associated with the MAMs via its proline-rich motif[24,25]. ATAD3A can regulate mitochondrial dynamics and maintain mitochondrial DNA (mtDNA) stability[25–27]. MAMs are a specialized subdomain of the ER with lipid raft features and rich in cholesterol and sphingomyelin[28]. Because of its unique localization on the MAMs, ATAD3A may regulate cholesterol trafficking through an unknown mechanism[26]. While global knockout of *ATAD3A* is embryonic lethal[29], selective loss of ATAD3A in mouse skeletal muscle disrupts mtDNA integrity and impairs cholesterol trafficking[30]. Thus, by connecting two subcellular organelles (the mitochondria and ER) via the MAMs, ATAD3A simultaneously regulates mitochondrial structure integrity and cholesterol metabolism. The dysregulation of both these processes is observed in the early stage of AD. Patients deficient in ATAD3A develop neurodegenerative conditions associated with axonal neuropathy[31], elevated free cholesterol, decreased expression of genes involved in cholesterol metabolism[26], and spastic paraplegia[32]. More recently, we reported that in the fatal and inherited neurodegenerative condition of Huntington's disease (HD), ATAD3A oligomerizes and accumulates at the contact sites of mitochondria and induces mitochondrial fragmentation, mitochondrial genome instability, and bioenergetic failure[27]. Moreover, blocking ATAD3A oligomerization by DA1, a peptide inhibitor, reduces HD pathology in various HD models[27]. Thus, ATAD3A may play an important role in the initiation and progression of neurodegeneration. However, whether ATAD3A is activated in AD and its exact roles in MAM hyperconnectivity and cholesterol disturbance underlying AD are unknown.

In this study, we reported that ATAD3A oligomerization increased at the MAMs in various AD disease models and the postmortem brains of AD patients. This aberrant oligomerization of ATAD3A induced AD-like hyperconnectivity of MAMs and impaired neuronal cholesterol turnover by inhibiting *CYP46A1* (Cytochrome P450 Family 46 Subfamily A Member 1) gene expression, which, in turn, promoted APP processing and synaptic loss. Notably, suppression of ATAD3A oligomerization by either heterozygous knockout or pharmacological inhibition in AD mice enhanced MAM integrity and cholesterol metabolism, suppressed APP processing, mitigated synaptic loss, and ultimately reduced AD-associated neuropathology and cognitive deficits. Thus, our results revealed that ATAD3A acts as a signaling node regulating MAM integrity to maintain cholesterol homeostasis and neuronal functions. Our findings also highlighted a potential therapeutic strategy for slowing AD progression by manipulating aberrant ATAD3A oligomerization.

## Results

**ATAD3A oligomerization increases in AD models**. To investigate the molecular involvement of ATAD3A in AD, we first carried out a computational analysis on the priority of ATAD3A in AD phenotypes, genes, and pathways by performing a virtual screening of a total of 10,072 prioritized disease phenotypes and 23,499 prioritized genes. We prioritized biomedical entities using a context-sensitive network-based ranking algorithm. The data mining showed that *ATAD3A* was closely associated with AD-specific phenotypes and AD-associated genes, ranking in the top 20.82% and 14.49%, respectively, which were significantly higher than random ranking ($p < 0.0001$; Supplementary Fig. 1a–c). The top-ranked pathways for ATAD3A were related to protein metabolism, gene transcription, immune response, and neurodegeneration (Supplementary Fig. 1d). These data suggested that ATAD3A could be involved in the development of AD pathology.

We recently reported that ATAD3A forms oligomers (mainly dimers) under pathological conditions, showing a gain-of-function that promotes neuropathology in HD models[27]. To determine the change in ATAD3A in AD, we first assessed

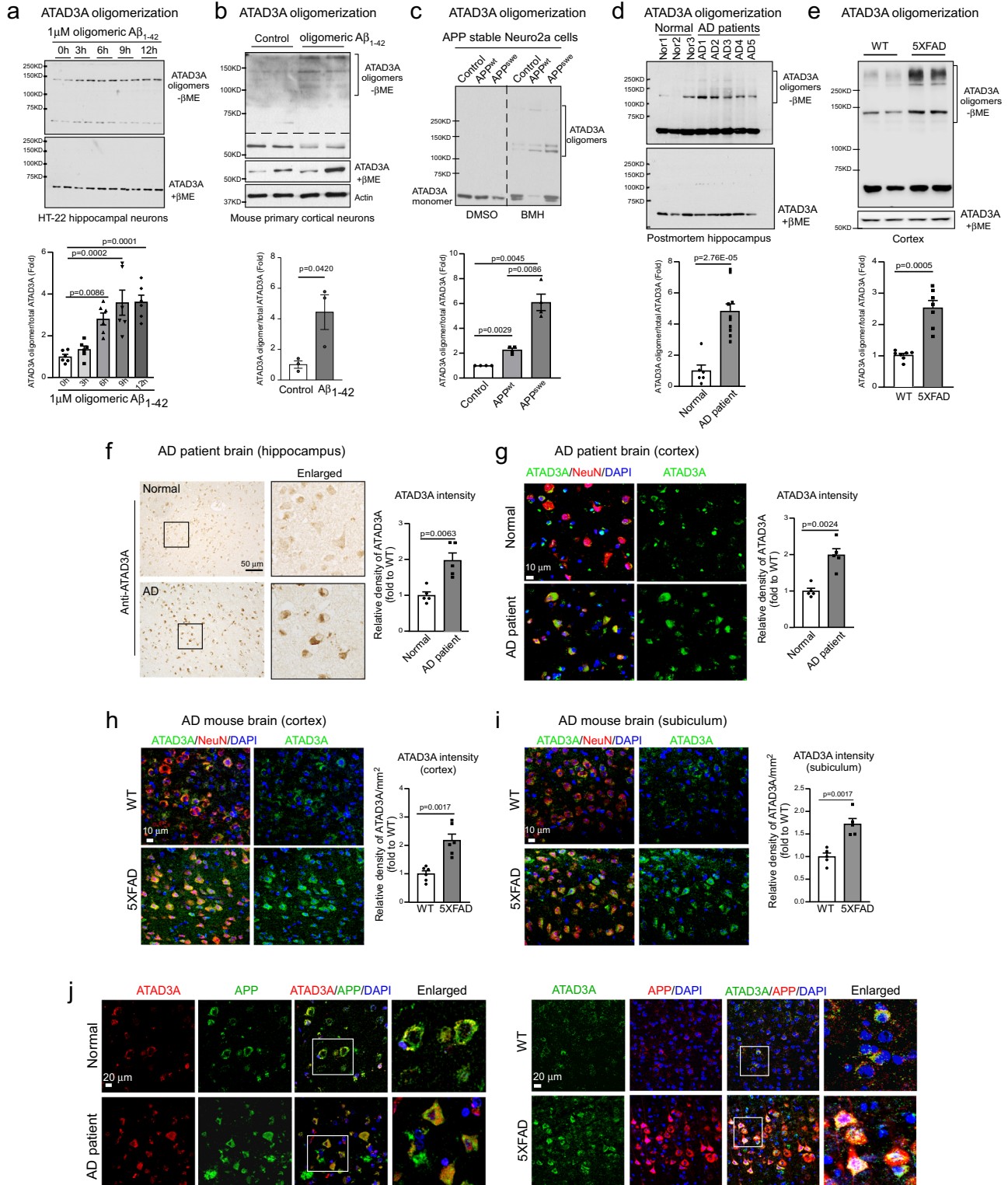

ATAD3A oligomerization in various AD experimental models. Under non-reducing conditions (i.e., in the absence of β-mercaptoethanol, β-ME), the levels of ATAD3A oligomers increased in immortalized mouse hippocampal HT-22 neurons and Neuro2a neuroblastoma cells exposed to oligomeric $A\beta_{1-42}$ peptide in a time- and dose-dependent manner (Fig. 1a, Supplementary Fig. 2a), and in toxic Aβ-treated mouse primary cortical neurons (Fig. 1b). In parallel, we confirmed an enhancement of ATAD3A oligomers in stable APP wildtype

(APP$^{wt}$)- and APP Swedish mutant (APP$^{swe}$)-expressing Neuro2a cells in the presence of a chemical cross-linker (bismaleimido-hexane, BMH), with a greater increase observed in the APP$^{swe}$-expressing cells (Fig. 1c). Consistently, ATAD3A oligomers increased in the total protein lysates from the postmortem hippocampus of AD patients under non-reducing conditions (Fig. 1d). ATAD3A oligomers were also elevated in the cortex, hippocampus, and thalamus of 5XFAD AD mice, but not in other brain regions (Fig. 1e, Supplementary Fig. 2c), consistent with

**Fig. 1 Aberrant ATAD3A oligomerization in AD models. a** HT-22 neuronal cells were treated with oligomeric $A\beta_{1-42}$ peptides (1 µM). $n = 6$ independent experiments, One-way ANOVA with Tukey's multiple comparisons test. **b** Primary mouse cortical neurons were treated with $A\beta_{1-42}$ peptides (1 µM, 12 h). ATAD3A protein levels were determined by western blotting (WB) with anti-ATAD3A antibody in the presence or absence of β-mercaptoethanol (βME). $n = 3$ independent experiments. **c** Stable APP^wt- and APP^swe-expressing Neuro2a cells were treated with the crosslinker BMH (1 mM) or DMSO control for 20 mins and then subjected to WB with an anti-ATAD3A antibody. $n = 4$ independent experiments. **d** ATAD3A oligomers were analyzed under non-reducing conditions in total lysates from postmortem hippocampus from AD patients by WB. Normal subjects (Nor): $n = 6$; AD patients: $n = 10$. **e** Total lysates from the cortex of 3-month-old 5XFAD mice or age-matched WT mice were analyzed by WB in the presence or absence of β-ME ($n = 7$ mice/group). Representative blots from at least three independent experiments are shown in **a–e**. The histograms in **a–e** show the density of ATAD3A oligomers relative to total ATAD3A levels in the presence of β-ME. **f** Postmortem hippocampus sections from normal subjects and AD patients were stained with anti-ATAD3A antibodies. The intensity of the ATAD3A staining was quantified ($n = 5$ individuals/group). **g** Postmortem cortex sections from normal subjects and AD patients were stained with anti-ATAD3A and anti-NeuN antibodies ($n = 5$ individuals/group). The intensity of ATAD3A staining in NeuN$^+$ cells was quantified. DAPI was used to label nuclei. Brain sections from 3-month-old WT and 5XFAD mice were stained with anti-ATAD3A and anti-NeuN antibodies. The ATAD3A immunodensity in NeuN$^+$ cells in the **h** cortex ($n = 6$ mice/group) or **i** subiculum ($n = 5$ mice/group) was quantified. **j** Postmortem cortex sections from normal subjects and AD patients (left) and brain sections from 3-month-old WT and 5XFAD mice (right) were stained with anti-ATAD3A and anti-APP antibodies. Human subject information is shown in Supplementary Fig. 2b. The data are presented as the mean ± SEM. Data in **b–i** were compared with the unpaired Student's $t$ test (two-tailed).

region-specific Aβ aggregation and human APP expression[33]. There was no change in ATAD3A ATPase activity in the cortex of 5XFAD mice relative to wildtype (WT) mice (Supplementary Fig. 2d). Overexpression of ATAD3A ATPase dead mutant ATAD3A-K358E-Flag[32] also had no effects on ATAD3A oligomerization (Supplementary Fig. 2e). Thus, ATAD3A oligomer formation is not associated with enzyme activity of the protein.

Immunohistochemical analysis revealed a higher ATAD3A staining in the postmortem hippocampus of AD patients than in normal subjects (Fig. 1f, Supplementary Fig. 2g). Moreover, we observed a significant increase in ATAD3A immunodensity in neurons immunopositive for anti-NeuN antibodies in the postmortem cortex of AD patients compared to normal subjects (Fig. 1g). The increased ATAD3A immunodensity in NeuN-immunopositive cells was consistently observed in cortical layer IV–V, the subiculum, and the hippocampus of 3-month-old 5XFAD AD mouse brains (Fig. 1h, i, Supplementary Fig. 2h). In addition, ATAD3A was enriched in APP-immunopositive cells of the postmortem cortex of AD patients and mice (Fig. 1j). The mRNA and total protein levels of ATAD3A were comparable in 3-month-old WT and 5XFAD mouse brains (Supplementary Fig. 2i, j). Thus, the elevated immunodensity of ATAD3A in AD patients and mouse brains is likely due to increased ATAD3A oligomerization, consistent with our previous observation[27]. Collectively, our data demonstrate an aberrant increase in ATAD3A oligomerization during the manifestation of AD, which supports our computational analysis results.

**ATAD3A accumulates at the MAMs and impairs MAM integrity in AD models.** We and others previously showed that ATAD3A localizes to mitochondria-associated contact sites and may be enriched in the MAMs[27,34]. In the present study, mitochondrial sub-compartmental fractionation from mouse brains revealed ATAD3A enrichment in the MAM fractions; ATAD3A was present in the same mitochondrial fractions as VDAC and SigmaR1, two proteins that have been localized to the MAMs[35] (Supplementary Fig. 3a). Notably, the distribution of ATAD3A to the MAM fraction was significantly enhanced in 5XFAD mice compared with WT mice (Supplementary Fig. 3a). There was also a significant increase in ATAD3A on the MAM fractions of Neuro2a cells treated with oligomeric $A\beta_{1-42}$ (Supplementary Fig. 3a). The physiological contact distance between the ER and mitochondria ranges between 10 and 30 nm[3,36], which allows the use of in situ proximity ligation assay (PLA) to assess ER-mitochondria tethering and localization of proteins on the MAMs. We observed a twofold increase in the number of PLA-positive puncta in 5XFAD

mouse cortex (Fig. 2a) and postmortem AD patient cortex (Fig. 2b) relative to control samples after staining brain sections with anti-SigmaR1 and anti-VDAC antibodies. The sizes of the PLA-positive puncta in these 5XFAD mice and AD patient brains were also larger than those of the control groups (Fig. 2a, b). In addition, high-resolution microscopy demonstrated a higher co-localization between IP3R3 and VDAC in HT-22 cells exposed to oligomeric $A\beta_{1-42}$ peptide (Supplementary Fig. 3b). These data suggest enhanced MAM tethering in AD models, which is in agreement with the hyperconnectivity of MAMs in AD[10,35]. Notably, we observed an approximately twofold increase in the number of PLA-positive puncta in the postmortem cortex of both 5XFAD mice and AD patients after staining with anti-ATAD3A and anti-FACL4 (a MAM marker) antibodies (Fig. 2a, b). The size of the PLA-positive puncta between ATAD3A and FACL4 in the 5XFAD AD mice and postmortem brains from AD patients was also significantly increased compared to the controls (Fig. 2a, b). These results demonstrated ATAD3A accumulation at the MAMs in the brains of AD patients and mice. Similarly, in HT-22 cells, treatment with oligomeric $A\beta_{1-42}$ peptide increased both the number and size of PLA-positive puncta when the cells were stained with anti-ATAD3A and anti-FACL4 antibodies (Fig. 2c). The PLA-positive signals were not observed in HT-22 cells stained for ATAD3A and cytochrome c (a mitochondrial intermembrane space protein), IP3R3 (a MAM protein) or SigmaR1 and mtCO1 (a mitochondrial inner membrane protein), or FACL4 and mtCO2 (a mitochondrial inner membrane protein) (Supplementary Fig. 3c), indicating the specificity of the PLA-positive puncta associated with ATAD3A and FACL4.

Based on our observations of ATAD3A oligomerization and accumulation at MAMs in various AD models, we determined the impact of aberrant ATAD3A oligomerization on ER-mitochondria tethering, a marker of MAM integrity and activity[11]. We knocked down ATAD3A in HT-22 cells using lentiviral *ATAD3A* shRNAs or treated the cells with DA1 peptide that we developed to block ATAD3A oligomerization[27]. In the presence of oligomeric $A\beta_{1-42}$ peptides, either ATAD3A down-regulation or DA1 treatment significantly reduced the number of PLA-positive puncta in Aβ-treated HT-22 cells stained with anti-IP3R3 and anti-VDAC antibodies or anti-SigmaR1 and anti-VDAC compared to control groups (Fig. 2d, Supplementary Fig. 3d). DA1 treatment also reduced Aβ-induced mitochondrial fragmentation (Supplementary Fig. 3e). Consistent with previous studies[11], oligomeric $A\beta_{1-42}$ increased IP3R3 and FACL4 protein levels, which was abolished by ATAD3A knockdown (Fig. 2e). Thus, increased ATAD3A oligomerization is required for AD-associated MAM hyperconnectivity.

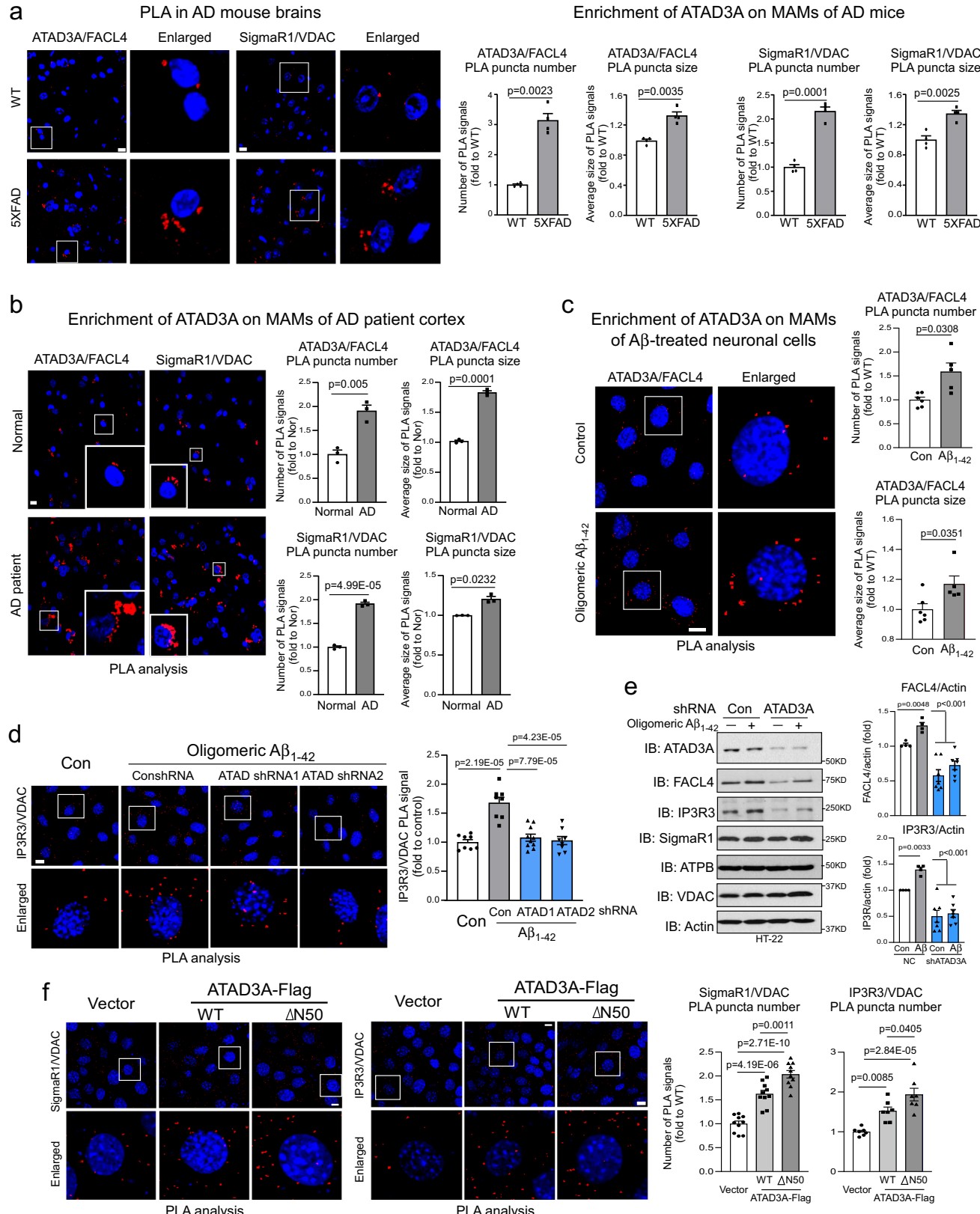

We previously demonstrated that a truncated ATAD3A mutant in which the first 50 amino acids are removed (ATAD3A ΔN50-Flag) enhanced ATAD3A oligomerization[27]. Here, we transduced HT-22 cells with ATAD3A-WT-Flag or truncated mutant ATAD3A-ΔN50-Flag and evaluated MAM tethering. Overexpression of ATAD3A-WT-Flag and ATAD3A-ΔN50-Flag significantly enhanced the number of PLA-positive puncta when HT-22 cells were stained with anti-SigmaR1 and anti-VDAC antibodies or anti-IP3R3 and anti-VDAC antibodies, with a higher number of PLA-positive puncta observed in cells expressing ATAD3A-ΔN50-Flag mutant (Fig. 2f). The expression of ATAD3A-WT-Flag or ATAD3A-ΔN50-Flag did not alter the

**Fig. 2 ATAD3A oligomerization impairs MAM integrity under AD conditions. a** Brain sections from 3-month-old WT and 5XFAD mice ($n = 4$ mice/group) and **b** postmortem cortex sections from normal subjects (Nor) and AD patients ($n = 3$ subjects/group) were stained with anti-SigmaR1 and anti-VADC antibodies or anti-ATAD3A and anti-FACL4 antibodies. Human subject information is presented in Supplementary Fig. 2b. The In situ Duolink proximity ligation assay (PLA) was performed. Histogram: the number and size of PLA-positive puncta (red). The number and size of PLA puncta signals were quantified from six separate fields of each sample. **c** HT-22 cells were treated with oligomeric $A\beta_{1-42}$ peptides ($5\,\mu M$) for 18 h. Control (Con): $n = 6$; $A\beta_{1-42}$: $n = 5$. **d** HT-22 cells were infected with control or *ATAD3A* shRNA lentivirus and then treated with oligomeric $A\beta_{1-42}$ peptides ($5\,\mu M$, 18 h). Cells were stained with the indicated antibodies and subjected to PLA analysis. Histogram: the number and size of PLA-positive puncta (red). $n = 10$ for sh*ATAD3A*-1/$A\beta_{1-42}$ group, $n = 8$ for the rest of three groups. At least 200 cells/group were analyzed. **e** Total protein lysates from the indicated groups were analyzed by WB. Histograms: the densities of FACL4 and IP3R3 relative to actin. $n = 4$ for Control group (NC) and $n = 7$ for sh*ATAD3A* group. **f** HT-22 cells were transfected with the indicated plasmids for 48 h. Cells were stained with the indicated antibodies and then subjected to PLA analysis. Histogram: the number and size of PLA-positive puncta (red). $n = 10$ for SigmaR1/VDAC group and $n = 7$ for IP3R3/VADAC group. At least 200 cells/group were analyzed. Scale bar: 10 μm. The data are presented as the mean ± SEM. Representative images and blots from at least three independent experiments are shown. The data in **a–c** were compared by the unpaired Student's $t$-test (two-tailed), and the data in **d-f** were compared by one-way ANOVA with Tukey's multiple comparisons test.

levels of the MAM-related proteins (Supplementary Fig. 3f), nor ATAD3A ATPase activity (Supplementary Fig. 2f). In contrast, the expression of ATAD3A-K358E-Flag, an ATAD3A ATPase dead mutant, did not affect MAM tethering (Supplementary Fig. 3g), further supporting the notion that the biological effects of ATAD3A oligomers do not result from its enzymatic activity. Collectively, our results indicate that ATAD3A oligomerization and accumulation at MAMs could induce the hyperconnectivity of ER-mitochondria tethering, reminiscent of AD-like pathology.

**ATAD3A haploinsufficiency reduces cognitive deficits and AD pathology in 5XFAD AD mice.** Next, we determined whether suppressing aberrant ATAD3A oligomerization affected AD-associated neuropathology and behavioral deficits in mice. *ATAD3A* homozygous knockout mice are embryonically lethal, but heterozygous knockout mice are normal and fertile[29,37]. Thus, we knocked out one *ATAD3A* allele from *ATAD3A*$^{fl/fl}$ mice by expressing CMV recombinase, which deleted loxP-flanked genes in all tissues (hereafter referred to as CMV; *ATAD3A*$^{fl/+}$). We then generated double-mutant 5XFAD$^{het}$; CMV; *ATAD3A*$^{fl/+}$ mice by crossing 5XFAD heterozygous mice with CMV; *ATAD3A*$^{fl/+}$ mice (Supplementary Fig. 4a). CMV; *ATAD3A*$^{fl/+}$ mice and 5XFAD$^{het}$; CMV;*ATAD3A*$^{fl/+}$ double-mutant mice were born at the expected Mendelian ratio and indistinguishable from WT and 5XFAD littermates, suggesting a lack of overt developmental deficits. No significant differences in body weight were observed between the four genotypes (Supplementary Fig. 4b). Western blot analysis revealed that total ATAD3A protein levels were lower in both the cortex and hippocampus of CMV; *ATAD3A*$^{fl/+}$ and 5XFAD$^{het}$; CMV; *ATAD3A*$^{fl/+}$ mice than in WT littermates and 5XFAD$^{het}$; *ATAD3A*$^{+/+}$ mice (Fig. 3a). The levels of proteins from the mitochondrial subcompartments (VDAC and Tom20, the outer membrane proteins; ClpP, the matrix protein; ATPB, the inner membrane protein) were comparable in all four mouse genotypes (Fig. 3a), suggesting that heterozygous knockout of *ATAD3A* did not alter mitochondrial mass. In addition, immunofluorescence staining confirmed ATAD3A downregulation in CMV; *ATAD3A*$^{fl/+}$ and 5XFAD$^{het}$; CMV; *ATAD3A*$^{fl/+}$ mice (Supplementary Fig. 4a). These data confirmed specific, heterozygous *ATAD3A* knockout in mice. Consistent with our findings, ATAD3A oligomers in 3-month-old 5XFAD$^{het}$; *ATAD3A*$^{+/+}$ mice were significantly higher than that in age-matched WT littermates. Notably, the level of ATAD3A oligomers in age-matched 5XFAD$^{het}$; CMV; *ATAD3A*$^{fl/+}$ mice returned to the levels observed in WT littermates (Fig. 3b). Thus, removing one copy of the *ATAD3A* gene reduced ATAD3A oligomerization in AD mice.

To assess the spatial learning and memory of 5XFAD$^{het}$;CMV;*ATAD3A*$^{fl/+}$ mice, we performed Y-maze and Barnes maze tests with mice of all four genotypes. 5XFAD$^{het}$;*ATAD3A*$^{+/+}$

mice had decreased short-term cognitive ability as assessed by the Y-maze at 6 months of age. In contrast, age-matched 5XFAD$^{het}$; CMV;*ATAD3A*$^{fl/+}$ mice had an improved spontaneous alteration ratio in the Y-maze test, reaching levels similar to those observed with WT mice (Fig. 3c). During the day 5 and day 12 assessments of the Barnes maze test, 8-month-old 5XFAD$^{het}$;*ATAD3A*$^{+/+}$ mice took a longer time and made more errors finding the target escape box than WT and CMV;*ATAD3A*$^{fl/+}$ mice, indicating a cognitive decline. Age-matched 5XFAD$^{het}$;CMV;*ATAD3A*$^{fl/+}$ mice exhibited a significant reduction in the latency and number of errors on day 5, and improved cognitive activity remained at day 12 (Fig. 3d; Supplementary Fig. 4c). Consistent with a previous report[38], 5XFAD mice were hyperactive during the open field test. 5XFAD$^{het}$;CMV;*ATAD3A*$^{fl/+}$ mice showed a normalized total distance traveled in the open field test similar to that of the WT mice (Supplementary Fig. 4d). These data suggest that reduced ATAD3A oligomerization improved the spatial learning and long-term memory of 5XFAD AD mice.

We stained brain sections of mice from the four genotypes with anti-SigmaR1 and anti-VDAC or anti-ATAD3A and anti-FACL4 antibodies and then performed PLA to assess MAM integrity in vivo. We observed an increased number of PLA-positive puncta in the cortex of 8-month-old 5XFAD$^{het}$;*ATAD3A*$^{+/+}$ mice, which was reduced in the age-matched 5XFAD$^{het}$;CMV;*A-TAD3A*$^{fl/+}$ mice to levels similar to those observed in WT mice (Fig. 3e, Supplementary Fig. 4e), demonstrating normalization of MAM hyperconnectivity by the *ATAD3A* heterozygous knockout. To assess the amyloid aggregation featured in 5XFAD AD mice, we stained brain sections of 8-month-old mice representing the four genotypes with anti-6E10 antibody, which labels amyloid aggregation. There was an increased number of 6E10$^+$ amyloid depositions and a larger area covered by amyloid plaques in the CA1 and subiculum regions of the hippocampus and cortex from 5XFAD$^{het}$;*ATAD3A*$^{+/+}$ mice. This abnormal amyloid accumulation was reduced in age-matched 5XFAD$^{het}$;CMV;*ATAD3A*$^{fl/+}$ mice (Fig. 3f). Neuroinflammation is another pathological marker of AD. We showed that the immunodensities of Iba1 (a marker of microglia) and GFAP (a marker of astrocytes) were significantly reduced in the cortex of 5XFAD$^{het}$;CMV;*ATAD3A*$^{fl/+}$ mice compared to 5XFAD$^{het}$;*ATAD3A*$^{+/+}$ mice (Fig. 3g), indicating a reduction of AD-associated gliosis. These data demonstrate that genetic reduction of enhanced ATAD3A oligomerization reduced neuropathology in 5XFAD AD mice.

**Inhibition of ATAD3A oligomerization by DA1 is neuroprotective in AD models.** We recently developed a peptide-based ATAD3A inhibitor, DA1, which specifically binds to ATAD3A protein and suppresses its oligomerization under stress conditions

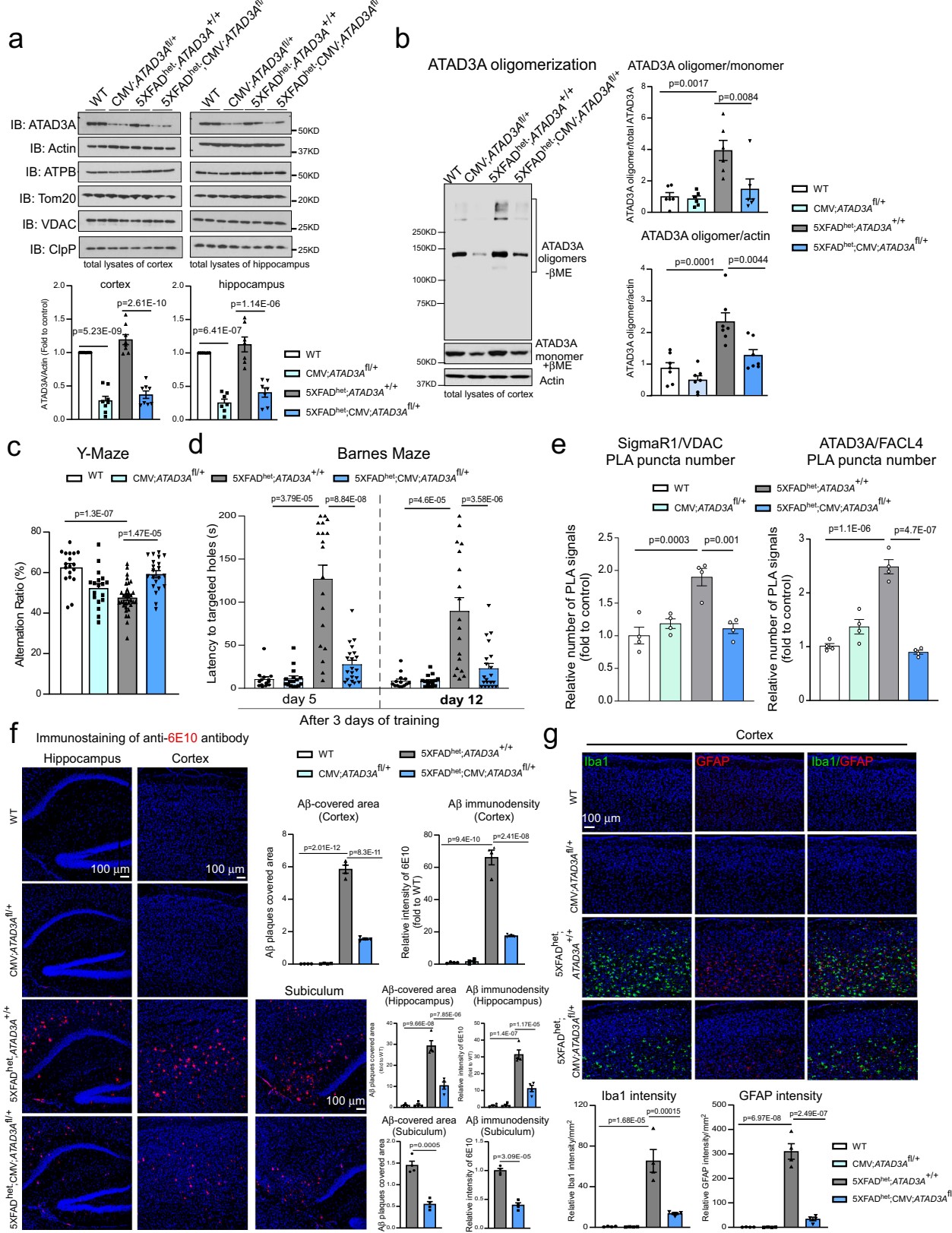

and in experimental models of HD[27]. DA1 can pass through the blood–brain barrier of mice and is tolerated by mice during long-term treatment ([27], Supplementary Fig. 5a–d). In addition, FITC-conjugated DA1 fluorescence signals significantly accumulated in the brains of WT mice one day after osmotic pump implantation (Supplementary Fig. 5e), confirming that DA1 administered

subcutaneously can enter mouse brains. In cultured HT-22 cells exposed to oligomeric $A\beta_{1-42}$ peptides, the DA1 peptide abolished ATAD3A oligomerization and elevated immunodensity signals (Supplementary Fig. 6a), validating the target. To test the in vivo efficacy of the DA1 peptide, we subcutaneously treated homozygous 5XFAD (5XFAD[homo]) mice with DA1 or control peptide

**Fig. 3 *ATAD3A* heterozygous knockout is neuroprotective in 5XFAD mice.** Total protein lysates were extracted from the cortex and hippocampus from 3-month-old WT, CMV; *ATAD3A*fl/+, 5XFADhet; *ATAD3A*+/+, and 5XFADhet; CMV; *ATAD3A*fl/+ mice. **a** WB was performed. Histogram: the density of ATAD3A relative to actin (n = 8 mice for cortex group, n = 7 mice for hippocampus group.). **b** ATAD3A oligomers were analyzed by WB under non-reducing conditions (βME: β-mercaptoethanol) (ATAD3A oligomer/monomer: n = 6 mice/group; ATAD3A oligomer/actin: n = 7 mice/group). Histogram: the density of ATAD3A oligomers relative to total ATAD3A protein levels or actin. **c** Short-term cognitive activity was assessed in 6-month-old mice of the indicated genotypes using the Y-maze (n = 18 mice for WT, n = 18 mice for CMV;*ATAD3A*fl/+, n = 33 mice for 5XFADhet;*ATAD3A*+/+, and n = 21 mice for 5XFADhet;CMV;*ATAD3A*fl/+ group). **d** Long-term cognitive activity was evaluated in 8-month-old mice of the indicated genotypes using the Barnes maze test (n = 14 mice for WT, n = 16 mice for CMV;*ATAD3A*fl/+, n = 19/18 mice for 5XFADhet; *ATAD3A*+/+ in day 5/12 group, and n = 21 mice for 5XFADhet; CMV; *ATAD3A*fl/+ group). Brain sections were prepared from 8-month-old WT, CMV;*ATAD3A*fl/+, 5XFADhet;*ATAD3A*+/+, and 5XFADhet;CMV;*ATAD3A*fl/+ mice (n = 4 mice/group). **e** Brain sections were stained with anti-SigmaR1 and anti-VDAC or anti-ATAD3A and anti-FACL4 antibodies and then subjected to PLA analysis (n = 4 mice/group). Histogram: the number of PLA-positive puncta (red). **f** Brain sections were stained with anti-6E10 antibody (red) to label amyloid deposits (n = 4 mice/group). The area covered by 6E10+ Aβ plaques and the immunodensity of 6E10 in the cortex, hippocampus, and subiculum were quantified from three separate fields of each mice. **g** Brain sections were stained with anti-Iba1 (green) and anti-GFAP (red) antibodies (n = 4 mice/group). The intensities of Iba1 and GFAP were quantified from three separate fields of each mice. Scale bar: 100 μm. Representative images and blots from at least three independent experiments are shown. All data are presented as the mean ± SEM and were compared by one-way ANOVA with Tukey's multiple comparisons test.

TAT (1 mg/kg/day) using an Alzet mini-pump from the age of 1.5 to 9 months (Supplementary Fig. 6b). Compared with 5XFADhet, 5XFADhomo mice develop amyloid pathology much more rapidly, together with broader neurological phenotypes, and lack gene-dosage effects[39], making them more suitable for assessing the efficacy of DA1 treatment. Treatment with DA1 not only abolished aberrant increase in ATAD3A oligomerization but also reduced enhanced ATAD3A immunodensity in 5XFADhomo mice (Fig. 4a, Supplementary Fig. 6a), confirming the inhibitory effect of DA1 in vivo. Moreover, treatment with DA1 suppressed MAM hyperconnectivity in the AD mice, as demonstrated by the reduced number of PLA-positive puncta in 6-month-old 5XFAD AD mouse brains following staining with anti-SigmaR1 and anti-VDAC or anti-ATAD3A and anti-FACL4 antibodies (Fig. 4b, Supplementary Fig. 6c). Notably, sustained DA1 treatment significantly improved the performance of 5XFAD mice in the Y-maze test at 6 months of age (Fig. 4c) and the Barnes maze test at 8 months of age (Fig. 4d; Supplementary Fig. 6d) compared to age-matched 5XFAD mice treated with the control peptide. In addition, DA1 treatment enhanced the nest-building ability of 5XFADhomo mice at 8.5 months of age, which was used as a complementary behavior assay because it is sensitive to spatial memory and hippocampus neuronal lesion in AD (Supplementary Fig. 6e). DA1 treatment also normalized the total traveled distance of 6-month-old 5XFADhomo mice relative to WT mice (Supplementary Fig. 6f). The body weight of mice was comparable between the TAT- and DA1-treated WT or 5XFAD mice (Supplementary Fig. 6g). Importantly, sustained treatment with DA1 (seven months) had no observable effects on the behavioral status or body weight of WT mice (Fig. 4, Supplementary Fig. 6), suggesting a lack of long-term toxicity.

Immunohistochemistry showed that treatment with DA1 reduced the Aβ-covered area and Aβ immunodensity in the cortex of 6-month-old 5XFADhomo mice (Fig. 4e). The treatment also abolished Iba1+ and GFAP+ immunoreactivity in the cortex and hippocampus of 5XFADhomo mice (Fig. 4f, Supplementary Fig. 6h), indicating inhibition of neuroinflammation. Neuronal loss has been observed in the hippocampus and cortical layer V in 5XFADhomo AD mice concomitant with amyloid aggregation and neuroinflammation[40,41]. We stained brain sections of 6-month-old 5XFADhomo mice with the Fluoro-Jade C (FJC) fluorescent probe that selectively binds to degenerating neurons[42] and anti-NeuN antibody. We observed FJC-positive fluorescence signals in NeuN+ cells in the CA1 region of the hippocampus and cortex, indicating ongoing neuronal loss (Fig. 4g). Treatment of 5XFADhomo mice with DA1 reduced the extent of neuronal degeneration. Indeed, the density of FJC fluorescence signals was decreased by more than 70% compared to 5XFADhomo mice

treated with control peptide TAT (Fig. 4g). Furthermore, sustained DA1 treatment did not elicit neuroinflammatory and neurodegenerative responses in WT mice (Fig. 4; Supplementary Fig. 6). Therefore, inhibition of ATAD3A oligomerization by DA1 reduced the AD neuropathology and cognitive deficits manifested in 5XFAD mice, consistent with our observations made in heterozygous *ATAD3A* knockout 5XFAD mice (Fig. 3; Supplementary Fig. 4).

**Aberrant ATAD3A oligomerization suppresses CYP46A1-mediated brain cholesterol turnover in AD models.** To investigate the mechanisms by which ATAD3A oligomerization mediates AD-associated neuropathology, we carried out unbiased label-free proteomic analysis on the brain tissue of 5XFADhet;CMV;*ATAD3A*fl/+ mice. We harvested the brain cortex from mice at 3 months of age, the age at which mice showed enhanced ATAD3A oligomerization but did not exhibit obvious amyloid accumulation or cognitive deficits. Among the 2639 proteins identified in the cortical tissue of 5XFADhet;CMV;*ATAD3A*fl/+, we focused on proteins that were altered in 5XFADhet;*ATAD3A*+/+ (i.e., >1.5-fold down-regulation or upregulation relative to WT mice) and simultaneously modified by heterozygous knockout of *ATAD3A* (i.e., >1.5-fold upregulation or downregulation relative to 5XFADhet;*ATAD3A*+/+; Fig. 5a). A total of 774 proteins that met these criteria (Fig. 5, marked with *) were subsequently used for pathway enrichment analysis. A graphical comparison of the KEGG analysis showed that the categories "metabolic pathway" and "Alzheimer's disease pathway" ranked as the top two protein enrichment pathways, in which proteins enriched for "lipid metabolic process" and "cholesterol metabolic process" were mostly affected, respectively (Fig. 5a, Supplementary Fig. 7a). Subsequent GO biological pathway analysis revealed that CYP46A1 was overlapped between the "lipid metabolic process" and "cholesterol metabolic process" (Fig. 5a, Supplementary Fig. 7a). Moreover, CYP46A1 ranked as the top candidate regulated by heterozygous *ATAD3A* knockout in AD mice. In particular, CYP46A1 was downregulated in 5XFADhet;*ATAD3A*+/+ mice relative to WT mice and restored in 5XFADhet;CMV;*ATAD3A*fl/+ mice (Fig. 5a, Supplementary Fig. 7a). CYP46A1 is a brain-specific enzyme that catalyzes cholesterol 24-hydroxylation, the main mechanism for cholesterol removal from the brain[43]. We hypothesize that ATAD3A oligomerization might cause AD-associated neuropathology and cognitive deficits by suppressing CYP46A1-mediated brain cholesterol metabolism.

The CYP46A1 immunodensity decreased in the cortex, hippocampus CA1, and subiculum of 5XFAD mouse brain, mainly in NeuN+ neuronal cells (Fig. 5b–e, Supplementary Fig. 7b), which is consistent with previous findings[44]. In contrast, the intensity of CYP46A1 staining in NeuN+ neurons was

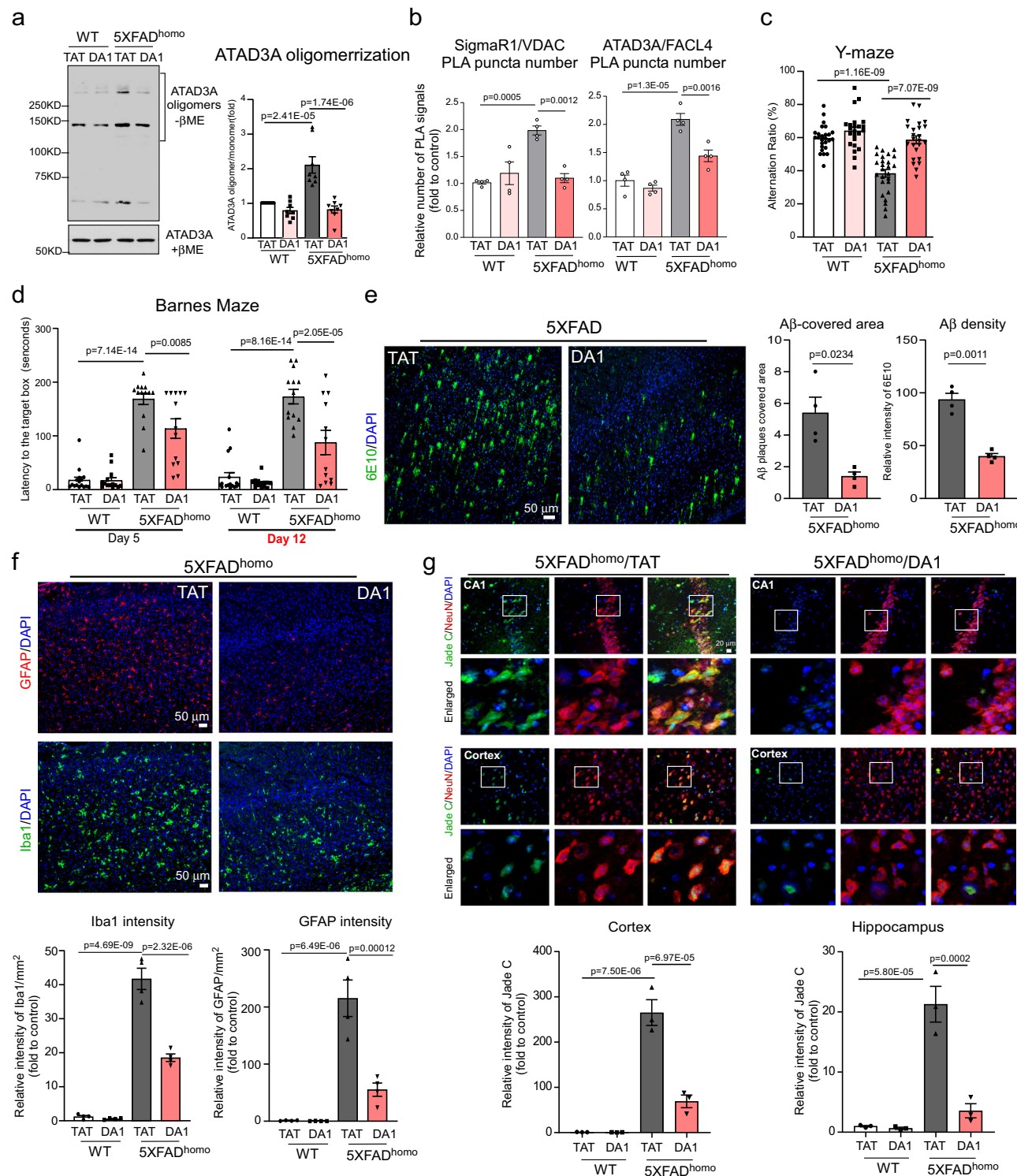

significantly elevated in both 5XFAD[het];CMV;*ATAD3A*[fl/+] mice and 5XFAD[homo] mice treated with DA1 peptides (Fig. 5b–e, Supplementary Fig. 7b). The decreased protein level of CYP46A1 in 5XFAD[het] mouse brains was also restored by heterozygous knockout of *ATAD3A* (Supplementary Fig. 7b). These results validated our proteomic analysis that demonstrated that a reduction in ATAD3A oligomerization improved CYP46A1 levels in AD mice. CYP46A1 deficiency causes cholesterol accumulation in neurons due to an impairment in neuronal cholesterol turnover[45]. ELISA analysis showed that the total cholesterol content increased in the cortex of 5XFAD mice,

however, this effect was significantly reduced in heterozygous *ATAD3A* knockout 5XFAD mice (Fig. 5f). Moreover, DA1 reduced the cholesterol content in the cortex of 5XFAD[homo] mice compared to that of AD mice treated with control peptide (Fig. 5g). Filipin is a commonly used fluorescence probe to monitor cholesterol deposition in cells and brain tissue[46]. We observed a significant accumulation of filipin-bound cholesterol in 9-month-old 5XFAD[homo] mouse brains and stable APP-expressing Neuro2a cells relative to their WT counterparts. Furthermore, inhibition of ATAD3A oligomerization by DA1 reduced cholesterol deposition (Fig. 5h, Supplementary Fig. 7c).

**Fig. 4 Suppression of ATAD3A oligomerization by DA1 is neuroprotective in AD mice.** 5XFAD and age-matched WT mice were subcutaneously treated with either TAT or DA1 beginning at six weeks of age (1 mg/kg/day) using an Alzet pump (treatment timeline shown in Supplementary Fig. 5b). **a** Total protein lysates were harvested from 6-month-old mouse brains, and ATAD3A oligomerization was analyzed by WB under non-reducing conditions (βME: β-mercaptaethanol). Histogram: the density of ATAD3A oligomers relative to total ATAD3A protein levels ($n = 8$ mice/group). **b** Brain sections from 6-month-old mice were stained with anti-SigmaR1 and anti-VDAC or anti-ATAD3A and anti-FACL4 antibodies and then subjected to PLA analysis. Histogram: the number of PLA-positive puncta (red) ($n = 4$ mice/group). **c** The Y-maze test was administered to 6-month-old mice (WT/TAT and 5XFAD[homo]/TAT: $n = 26$ mice/group; $n = 23$ mice for the WT/DA1 group and $n = 24$ mice for the 5XFAD[homo]/DA1 group). **d** The Barnes maze test was administered to 8-month-old mice (WT/TAT: $n = 16$ mice/group; WT/DA1: $n = 14$ mice/group; 5XFAD[homo]/TAT: $n = 13$ mice/group; 5XFAD[homo]/DA1: $n = 13/12$ mice/group in day5/ day12 test). **e** Brain sections from 6-month-old mice of the indicated groups were stained with an anti-6E10 antibody. The area covered by 6E10$^+$ Aβ plaques and the immunodensity of 6E10 in cortex were quantified from three separate fields of each mice ($n = 4$ mice/group). **f** Brain sections from 6-month-old mice of the indicated groups were stained with anti-Iba1 (green) and anti-GFAP (red) antibodies. The intensity of Iba1 and GFAP was quantified from four separate fields of each mice ($n = 4$ mice/group). Scale bar: 50 μm. **g** Brain sections from 6-month-old mice were stained with Jade C probe (green) and anti-NeuN antibody (red). The Jade C intensity in NeuN+ cells in the cortex and hippocampus was quantified and shown in the histograms ($n = 3$ mice/group). Scale bar: 20 μm. Representative images and blots are at least three independent experiments are shown. All data are presented as the mean ± SEM and were compared by one-way ANOVA with Tukey's multiple comparisons test.

Conversion of brain cholesterol into 24(S)-hydroxycholesterol (24-OHC) represents the primary cholesterol elimination mechanism in the brain. The 24-OHC content has been used as a marker of brain cholesterol dysregulation in AD[47]. Unlike cholesterol, 24-OHC can cross the blood-brain barrier at a high rate. More than 90% of 24-OHC in plasma is processed from the brain, making plasma 24-OHC concentrations a useful marker to monitor brain cholesterol turnover and neuronal CYP46A1 activity[48]. We collected plasma from 9-month-old WT and 5XFAD[homo] mice treated with DA1 or control peptide TAT. The concentration of 24-OHC was decreased in the plasma of 5XFAD[homo] mice, reflecting the suppression of brain cholesterol elimination. Importantly, sustained treatment of 5XFAD[homo] mice with DA1 peptide restored the plasma concentrations of 24-OHC to levels similar to those measured in WT mice treated with control peptide (Fig. 5i), further supporting a role for ATAD3A oligomerization in inhibiting CYP46A1 in AD mouse brains. In parallel, we profiled brain sterols in heterozygous *ATAD3A* knockout 5XFAD and DA1-treated 5XFAD mouse cortex using gas chromatography-mass spectrometry (GC-MS). The levels of the sterols (e.g., lanosterol, zymosterol, desmosterol, and lathosterol) were comparable between all analyzed experimental groups (Supplementary Fig. 7d). Thus, ATAD3A oligomerization affects brain cholesterol levels in AD models by altering cholesterol metabolism, not biosynthesis.

**ATAD3A oligomerization inhibits CYP46A1 at the transcriptional level.** Next, we set out to determine how aberrant ATAD3A oligomerization affected CYP46A1 expression. Overexpression of ATAD3A-WT-Flag or ATAD3A-ΔN50-Flag, which enhances ATAD3A oligomerization strikingly decreased the *CYP46A1* mRNA and protein levels in Neuro2a cells (Fig. 5j; Supplementary Fig. 7e) but did not alter the protein levels of mitochondrial ATPB and VDAC or the MAM protein SigmaR1 (Supplementary Fig. 7e). ATAD3A-WT and ATAD3A-ΔN50 did not change the mRNA levels of *CYP51A1* and *HMGCS1*, genes involved in cholesterol metabolism and biosynthesis, respectively[49,50] (Supplementary Fig. 7f). In HT-22 cells exposed to oligomeric Aβ$_{1-42}$ peptide or stable APP-expressing Neuro2a cells, inhibition of ATAD3A oligomerization by DA1 significantly enhanced *CYP46A1* mRNA levels (Fig. 5k). In contrast, the *CYP51A1* mRNA levels were comparable between the different experimental groups under the same conditions (Supplementary Fig. 7g). ATAD3A protein levels and oligomerization were not affected by CYP46A1 overexpression or inhibition of CYP46A1 enzyme activity by voriconazole (Supplementary Fig. 7h, i). Thus, ATAD3A acts upstream of CYP46A1, and ATAD3A oligomerization selectively suppresses CYP46A1 at the transcriptional level.

Overexpression of ATAD3A-WT-Flag and ATAD3A-ΔN50-Flag variants in Neuro2a cells increased the total cholesterol content and decreased the 24-OHC content compared to cells expressing the control vector, with a worse extent observed in ATAD3A-ΔN50-Flag-expressing cells (Fig. 5l, m, Supplementary Fig. 7j). In contrast, DA1 treatment abolished ATAD3A-WT- or -ΔN50-induced cholesterol accumulation, and overexpression of ATAD3A-ΔCC-Flag variant, which blocks ATAD3A oligomer formation, showed a comparable effect on cholesterol content to that seen in control vector-expressing cells (Supplementary Fig. 7j). These data support a direct effect of ATAD3A oligomerization on cholesterol accumulation. Furthermore, CYP46A1 overexpression reduced the cholesterol content increase and corrected the 24-OHC level in both ATAD3A-WT-Flag and - ΔN50-Flag expressing cell lines (Fig. 5l, m). Similarly, treatment with efavirenz (EFV), an allosterical activator of CYP46A1, also abolished ATAD3A-WT- and -ΔN50-induced cholesterol accumulation (Supplementary Fig. 7k). Thus, ATAD3A oligomerization-induced perturbation of cholesterol homeostasis depends on CYP46A1. Among the 16 genes implicated in cholesterol metabolic pathways, *LDLR* and *ApoE* mRNA levels were significantly elevated in 5XFAD mice but attenuated in both 5XFAD[het];CMV;*ATAD3A*[fl/+] mice and DA1-treated 5xFAD[homo] mice (Supplementary Fig. 7l). These results are in line with the fact that 24-OHC is an endogenous agonist of nuclear liver X receptors (LXRs), which subsequently regulate LXR-targeted gene expression (e.g., *LDLR* and *ApoE*), balancing brain cholesterol homeostasis[51]. Thus, a reduction in *LDLR* and *ApoE* mRNA levels in AD mice by ATAD3A heterozygous knockout or DA1 treatment is consistent with enhanced CYP46A1 levels. Altogether, these data support our hypothesis that a loss of CYP46A1 is, at least in part, responsible for the ATAD3A oligomerization-induced neuronal cholesterol accumulation in AD models.

**ATAD3A oligomerization promotes APP processing in a CYP46A1-dependent manner.** Compensation for CYP46A1 deficiency in vivo reduces amyloid deposits and improves spatial memory in AD mice[52]. The mechanisms underlying the neuroprotection provided by CYP46A1 overexpression are thought to decrease the cholesterol content in the lipid rafts of membranes and reduce amyloidogenic APP processing[52]. In the postmortem cortex of AD patients, we observed enlarged and swollen lipid rafts that were immunopositive for anti-cholera toxin B pentamer (CTxB), a lipid raft marker. Moreover, CTxB-positive lipid rafts were colocalized with APP (Supplementary Fig. 8a), consistent with the notion that APP is enriched at lipid rafts for processing[53]. In 5XFAD mouse brains, the intensity of the CTxB-positive lipid rafts increased in cortical layer V and colocalized with increased APP protein expression (Supplementary Fig. 8b).

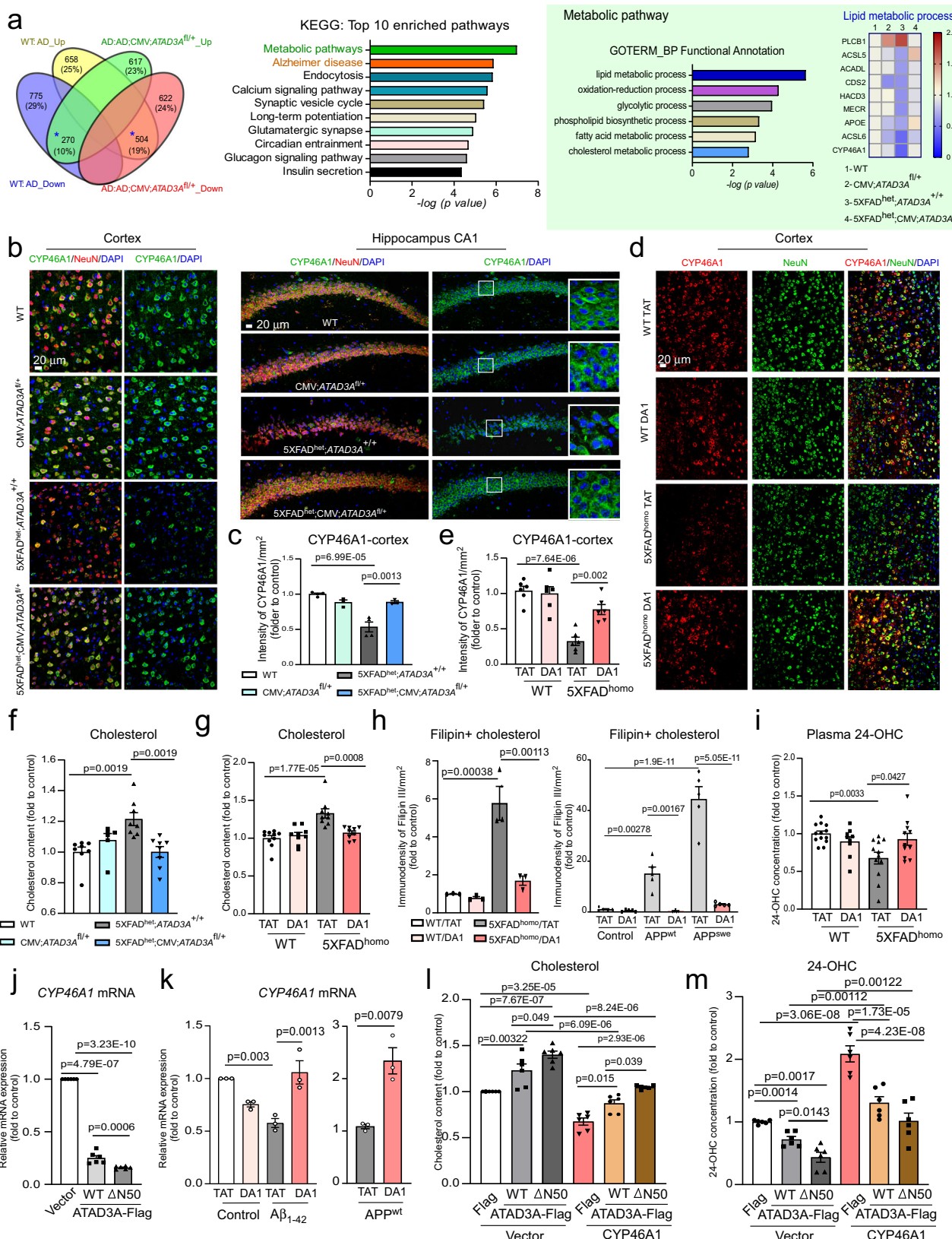

Heterozygous *ATAD3A* knockout in 5XFAD[het] mice or DA1-mediated suppression of ATAD3A oligomerization attenuated the CTxB immunodensity (Fig. 6a, b), suggesting the normalization of the lipid rafts. Because the ATAD3A intensity increased in APP-immunopositive cells in the brains of both AD patients and AD 5XFAD mice (Fig. 1j), we examined whether APP

processing was affected by aberrant ATAD3A oligomerization. Enhanced ATAD3A oligomerization mediated by overexpression of ATAD3A-WT-Flag or ATAD3A-ΔN50-Flag exacerbated the production of the C99 fragment, a pathological proteolytic product of APP present in stable APP[wt]-expressing Neuro2a cells. In contrast, blockage of ATAD3A oligomerization by either

**Fig. 5 ATAD3A oligomerization inhibits cholesterol turnover in AD models. a** The cortex of 3-month-old mice ($n = 3$ mice/group) was subjected to Label-free tandem mass spectrometry. Left: the number of proteins changed in mice. Middle: KEGG database analysis on proteins from the pool marked as *. Right: GO biological pathway analysis of the proteins enriched in the "metabolic pathway". Heat map: proteins involved in the "lipid metabolic process". Brain sections from **b** 8-month-old mice and **d** 6-month-old mice were stained with anti-CYP46A1 and anti-NeuN antibodies. The intensity of CYP46A1 in NeuN+ cells from mouse cortex was quantified from three separate fields per mouse in **c** (WT and 5XFAD^het;ATAD3A^+/+: $n = 4$; CMV;ATAD3A^fl/+ and 5XFAD^het;CMV;ATAD3A^fl/+: $n = 3$) and **e** ($n = 6$ mice/group). Scale bar: 20 μm. The total cholesterol content was measured in **f** the cortex from 8-month-old mice and **g** 9-month-old mice (WT: $n = 8$; CMV;ATAD3A^fl/+: $n = 6$; 5XFAD^het;ATAD3A^+/+: $n = 8$ and 5XFAD^het;CMV;ATAD3A^fl/+: $n = 8$. WT/TAT: $n = 10$; the rest of groups: $n = 9$). **h** Left: Brain sections from 9-month-old mice were stained with the filipin probe. The immunodensity of filipin+ cholesterol was quantified from three separate fields per mouse ($n = 3$ mice/group). Right: Stable APP^wt- and APP^swe-expressing Neuro2a cells were treated with DA1 and TAT (1 μM/day for 4 days) and stained with the filipin probe. The intensity of filipin+ cholesterol was quantified from five separate fields per sample ($n = 5$ independent biological experiments). Representative images are shown in Supplementary Fig. 6c. **i** Plasma 24-OHC levels from 9-month-old mice were measured (WT/TAT: $n = 13$; WT/DA1: $n = 9$; 5XFAD^homo/TAT: $n = 12$; 5XFAD^homo/DA1: $n = 11$). **j** CYP46A1 mRNA was measured by qPCR ($n = 6$). **k** Left: HT-22 cells were treated with TAT or DA1 (1 μM) and Aβ₁–₄₂ peptides (5 μM, 9 h). Right: Stable APP^wt-expressing Neuro2a cells were treated with TAT or DA1. CYP46A1 mRNA was measured by qPCR ($n = 3$). **l** Total cholesterol content was measured ($n = 6$). **m** The concentration of 24-OHC was measured ($n = 6$). All data are presented as mean ± SEM from at least three independent experiments and compared by the unpaired two-tailed Student's $t$ test (**k**-right) or one-way ANOVA with Tukey's multiple comparisons test (**c**–**m**).

expression of the ATAD3A-ΔCC-Flag variant or treatment with DA1 abolished the C99 production (Fig. 6c), indicative of a direct impact. DA1 treatment also abolished the C99 fragment in stable APP^wt- and APP^swe-expressing Neuro2a cells (Supplementary Fig. 8c). Furthermore, we observed a significant increase in C99 fragment levels in 5XFAD mouse brains, which was reduced by either heterozygous ATAD3A knockout or DA1 treatment (Fig. 6d). Thus, inhibition of ATAD3A oligomerization could reduce APP processing in AD models, consistent with reducing amyloid aggregation in both 5XFAD^het;CMV;ATAD3A^fl/+ mice and DA1-treated 5XFAD^homo mice (Figs. 3, 4).

As part of the intracellular lipid rafts, MAMs provide a crucial signal platform for APP processing. Recent proteomic analysis demonstrated that proteins involved in cholesterol metabolism and Aβ clearance (e.g., CYP46A1 and ABCG1) resided on the MAMs and were altered in the early presymptomatic stage of AD[54]. Like APP, CYP46A1 could be detected in the MAM fraction of mouse brains, although CYP46A1 was enriched in the ER (Supplementary Fig. 8d). PLA-positive puncta between CYP46A1 and VDAC was also observed in WT mouse brain (Supplementary Fig. 8e). Moreover, ATAD3A and CYP46A1 interacted in the brains of WT mice, whereas the interaction was decreased in 5XFAD mouse brains, most likely due to the loss of CYP46A1 in the AD mouse brain (Supplementary Fig. 8f). These data indicated that CYP46A1 present on the MAMs formed a complex with ATAD3A. To determine if the loss of CYP46A1 mediated ATAD3A oligomerization-induced APP processing, we overexpressed CYP46A1 or control vector in stable APP-expressing Neuro2a cells in the presence of DA1 peptide or control peptide TAT. Similar to the results observed with stable APP-expressing Neuro2a cells treated with DA1 peptide, overexpression of CYP46A1 alone abolished C99 fragment levels in stable APP^wt- and APP^swe-expressing Neuro2a cells (Fig. 6e), suggesting the inhibition of APP processing. DA1-mediated suppression of ATAD3A oligomerization followed by overexpression of CYP46A1 had no additive effects on C99 fragment levels (Fig. 6e). These results are consistent with our observation that EFV treatment abolished the ATAD3A-WT- or -ΔN50-induced increase in C99 production in stable APP^wt Neuro2a cells (Supplementary Fig. 8g). Collectively, our results suggest that in AD, ATAD3A cooperates with CYP46A1 to mediate APP processing, presumably at the MAMs.

**ATAD3A oligomerization causes synaptic loss in AD models.**
Synaptic loss is observed in AD pathology, and disruption of brain cholesterol metabolism has been shown to lead to synaptic loss and subsequent cognitive deficits[55]. We assessed whether aberrant ATAD3A oligomerization influenced synaptic

morphology by quantifying synaptic density using anti-synaptophysin (a presynaptic protein) and anti-PSD95 (a post-synaptic protein) co-staining. In primary mouse cortical neurons, overexpression of either Flag-tagged ATAD3A-WT or ATAD3A-ΔN50 decreased the colocalization of synaptophysin and PSD95 along the dendrites, reflecting a reduction in synaptic density, and increased neuronal cell death (Fig. 7a, Supplementary Fig. 9a). Moreover, the treatment of primary neurons with oligomeric Aβ₁–₄₂ peptide reduced synaptic density and induced cell death (Fig. 7b, c) These effects were corrected by either ATAD3A knockdown or DA1 treatment (Fig. 7b, c, Supplementary Fig. 9b). Furthermore, CYP46A1 overexpression alone increased synaptic density in primary neurons exposed to oligomeric Aβ₁–₄₂ peptide. Blocking ATAD3A oligomerization with DA1 followed by CYP46A1 overexpression did not have an additional effect on the number of synaptophysin-positive clusters along the MAP2+ dendrites compared to DA1 treatment or CYP46A1 over-expression alone (Fig. 7d). Similarly, enhancement of CYP46A1 by EFV treatment abolished ATAD3A-WT- or ATAD3A-ΔN50-induced synaptic loss in primary neurons (Supplementary Fig. 9c). These results further support the hypothesis that CYP46A1 mediates ATAD3A oligomerization-induced neuronal damage in AD by impairing cholesterol turnover.

Finally, we assessed the effects of ATAD3A oligomerization on synaptic proteins and synaptic morphology in AD mice. Western blot analysis showed significant reductions in synaptophysin and PSD95 protein levels in the cortex of 5XFAD mice; however, their levels were restored by either heterozygous ATAD3A knockout or DA1 treatment (Fig. 7e, f). The neuronal spine density assessed by Golgi–Cox staining was decreased in 9-month-old 5XFAD^homo mice. Notably, treatment with DA1 in age-matched AD mice increased the number of dendritic spines compared to 5XFAD mice treated with control peptide (Fig. 7g). Thus, suppression of ATAD3A oligomerization reduced synaptic loss in 5XFAD mice, indicating the potential for improving the cognitive activity of AD mice by genetic or pharmacological inhibition of ATAD3A oligomerization (Figs. 3, 4).

## Discussion

In this study, we demonstrated that pathological oligomerization of ATAD3A upregulated ER-mitochondrial connections, impaired cholesterol homeostasis, and promoted amyloid processing, leading to neurodegeneration in AD. Moreover, we discovered the ATAD3A-CYP46A1-APP signaling axis that mediates the development of AD pathology and cognitive deficits (Supplementary Fig. 9d). Therefore, our research findings provide insights into the pathogenesis of AD and demonstrate that

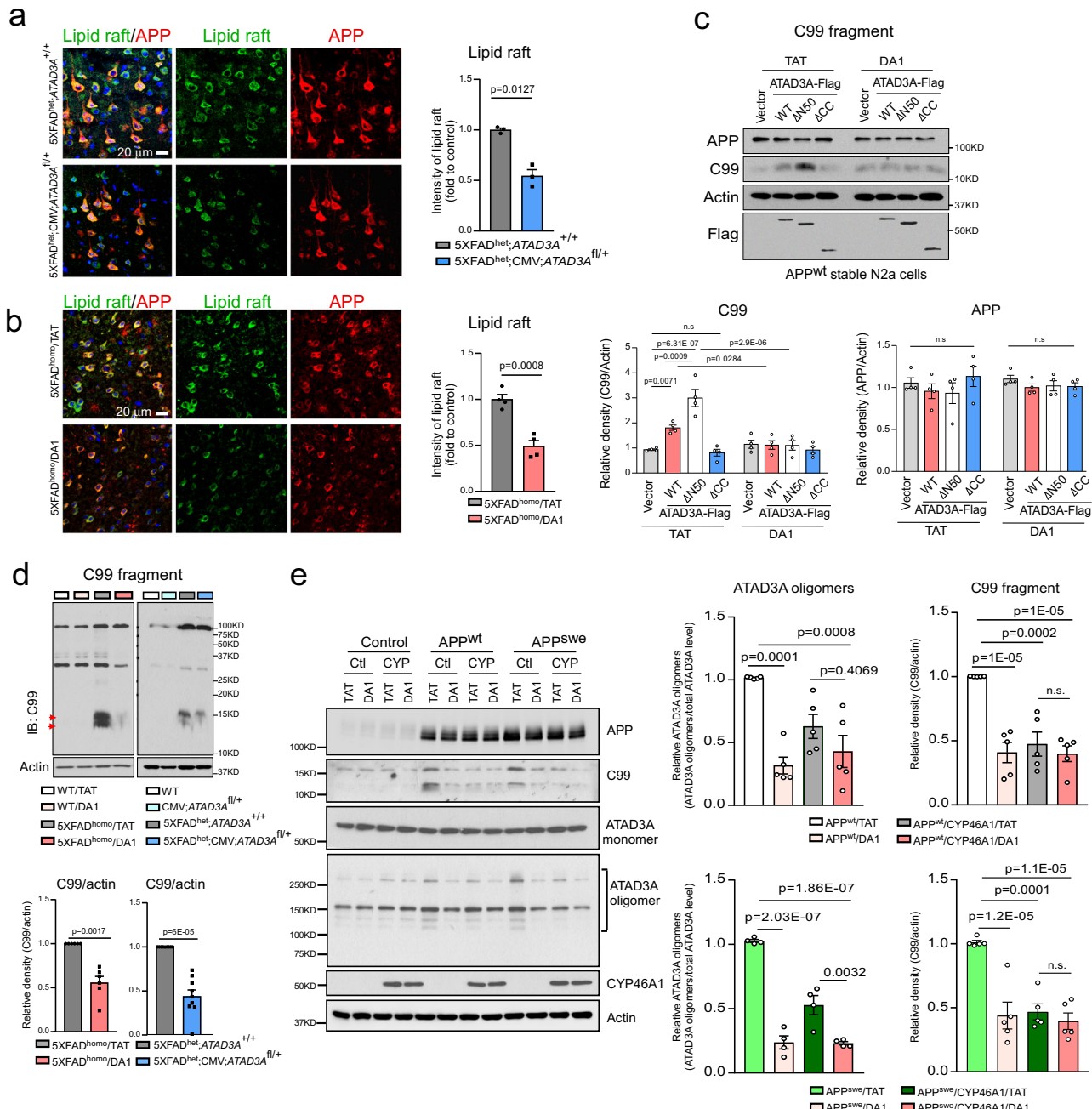

**Fig. 6 ATAD3A and CYP46A1 cooperate to promote APP processing. a** Brain sections from 3-month-old mice of the indicated genotypes were stained with anti-CTxB (green) and anti-APP (red) antibodies. The intensity of CTxB in the mouse cortex was quantified from three separate fields of each mouse and shown in the histogram ($n = 3$ mice/group). **b** Brain sections from 9-month-old mice of the indicated treatment groups were stained with anti-CTxB (green) and anti-APP (red) antibodies. The intensity of CTxB in the mouse cortex was quantified from three separate fields of each mouse and shown in the histogram ($n = 4$ mice/group). **c** Stable APP$^{wt}$-expressing Neuro2a (N2a) cells were transfected with ATAD3A-WT-Flag, ATAD3A-$\Delta$N50-Flag, ATAD3A-$\Delta$CC-Flag, or control vector for 24 h, followed by treatment with DA1 or control peptide TAT for 48 h (1 µM, each). WB was performed. Histogram: the densities of C99 and APP relative to actin ($n = 4$ independent experiments). **d** Total brain lysates were harvested from the cortex of 8-month-old 5XFAD$^{het}$;CMV;ATAD3A$^{fl/+}$ group or 9-month-old mice treated with DA1. The APP proteolytic product C99 was assessed by WB (Indicated by red arrow). Histogram: the density of C99 relative to actin ($n = 6$ mice for DA1 treated groups, $n = 9$ mice for 5XFAD$^{het}$;CMV;ATAD3A$^{fl/+}$ groups). **e** CYP46A1 was overexpressed in stable APP$^{wt}$- and APP$^{swe}$-expressing Neuro2a cells, which were then treated with DA1 or control peptide TAT (1 µM for 3 days). WB was performed. Ctl and CYP on the images indicate Control Vector and CYP46A1, respectively. The relative densities of the ATAD3A oligomers and C99 were quantified ($n = 5$). Representative images and blots from at least three independent experiments are shown. All data are presented as the mean ± SEM. The data in **a** and **b** were compared by the unpaired Student's $t$ test (two-tailed), and the data in **c**–**e** were compared by one-way ANOVA with Tukey's multiple comparisons test.

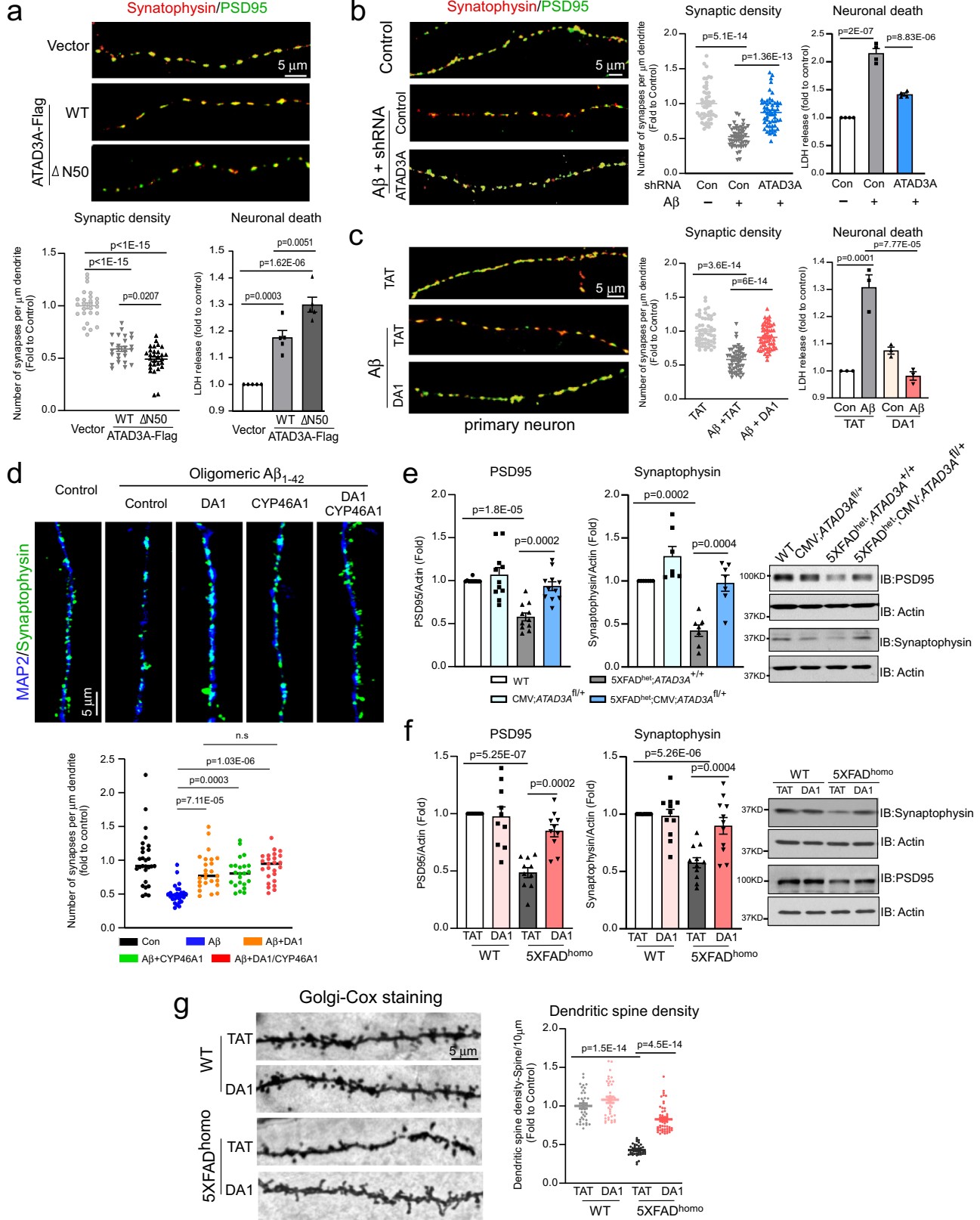

ATAD3A oligomerization is a potential therapeutic target for the treatment of AD and other neurological disorders associated with MAM hyperconnectivity and cholesterol disturbance.

Environmental and genetic factors involved in the disturbance of cholesterol metabolism have been suggested as risk factors for AD development[56]. Abnormal retention of brain cholesterol causes

increased Aβ production, secretion, and fibrillization and facilitates Aβ toxicity[57,58]. In turn, Aβ can modulate cholesterol homeostasis, establishing a vicious cycle between cholesterol accumulation and Aβ generation. CYP46A1 is a cholesterol degradation enzyme that converts cholesterol to 24-OHC by hydroxylation, which is the key process mediating brain cholesterol elimination and turnover[45].

**Fig. 7 ATAD3A oligomerization causes synaptic loss in AD models. a** Primary mouse cortical neurons (DIV 7 days) were transfected with the indicated plasmids for 48 h. **b** Primary mouse cortical neurons were infected with lentiviral *ATAD3A* or control (Con) shRNAs for 48 h and then treated with Aβ$_{1-42}$ peptides (1 μM, 12 h). **c** Primary mouse cortical neurons were treated with DA1 or TAT (1 μM) followed by treatment with Aβ$_{1-42}$ peptides (1 μM, 12 h). The neuronal cells were stained with anti-synaptophysin and anti-PSD95 antibodies. The synaptophysin$^+$PSD95$^+$ clusters along the dendrites were counted, and the number of synapses per micron of dendrites was quantified. $n = 26$–29 neurons in **a**, $n = 50$–54 neurons in **b** and $n = 54$–57 neurons in **c**. Cell death was measured by the release of LDH from at least three independent experiments, $n = 5$ in **a**, $n = 4$ (**b**), n = 3 in **c**. **d** Primary mouse cortical neurons were infected with CYP46A1 or control lentivirus (Con) for 72 h. The cells were then treated with DA1 or TAT (1 μM) followed by treatment with Aβ$_{1-42}$ (1 μM, 12 h). The neuronal cells were stained with anti-synaptophysin and anti-MAP2 antibodies. The synaptophysin$^+$ clusters along the MAP2$^+$ dendrites were counted, and the number of synapses per micron of dendrites was quantified. $n = 23$–28 neurons from at least three independent experiments. Representative images from at least three independent experiments are shown. Scale bar: 5 μm. **e** Total protein lysates were prepared from the cortex of 8-month-old mice ($n = 11$ mice/group for PSD95 analysis, $n = 7$ mice/group for Synaptophysin analysis.). **f** Total protein lysates were harvested from cortex of the 9-month-old mice ($n = 10$ mice/group for PSD95 analysis, $n = 11$ mice/group for Synaptophysin analysis.). WB was performed. Histogram: the densities of PSD95 and synaptophysin relative to actin. **g** Golgi-Cox staining of mouse brains was shown. The relative spine number per 10 μm dendrite was quantified ($n = 3$ mice/group, 37–54 neurons/group). Scale bar: 5 μm. All data are presented as the mean ± SEM and compared by one-way ANOVA with Tukey's multiple comparisons test.

*CYP46A1* polymorphisms are a risk factor for AD[59]. CYP46A1 deficiency causes brain cholesterol accumulation, amyloid accumulation and aggregation, and cognitive deficits[60]. Compensation for CYP46A1 deficiency in AD mice markedly reduces amyloid deposits and improves spatial memory[52]. However, the mechanism of CYP46A1 loss in AD remains unknown. In the present study, we discovered that ATAD3A oligomerization acts upstream of CYP46A1 and is a trigger for CYP46A1 deficiency. We established a molecular model that under basal condition, both ATAD3A and CYP46A1 reside on the MAMs where ATAD3A interacts with and stabilizes CYP46A1. Under AD-like pathological conditions, ATAD3A oligomerization inhibits CYP46A1 at the transcriptional level, leading to neuronal cholesterol accumulation, MAM hyperconnectivity, and, ultimately, synaptic loss. This hypothesis was supported by the observations that suppression of ATAD3A oligomerization by heterozygous knockout or pharmacological inhibition restored neuronal CYP46A1 levels, promoted brain cholesterol turnover, and normalized MAM tethering, resulting in a reduction in amyloidogenesis and improved cognitive ability in AD mice. Moreover, multiple lines of evidence from our study indicate that loss of CYP46A1 is the underlying mechanism for ATAD3A oligomerization-induced neuropathology. Thus, our research findings support the notion that ATAD3A and CYP46A1 synergistically regulate cholesterol metabolism and amyloidogenesis. Future studies of CYP46A1-overexpression in neurons of ATAD3A heterozygous knockout AD mice will further help to address the impact of neuronal CYP46A1 restoration on the MAM impairment, cholesterol metabolism dysregulation, and AD-associated neuropathology that are mediated by ATAD3A oligomerization.

Currently, the mechanism by which ATAD3A oligomerization suppresses *CYP46A1* gene expression remains unknown. There are several possibilities. *CYP46A1* mRNA expression could be regulated by oxidative stress[61]. ATAD3A oligomerization-induced oxidative stress[27] may indirectly suppress *CYP46A1* transcription. It is also possible that ATAD3A oligomerization under AD conditions disrupts the ATAD3A/CYP46A1 complex, resulting in cholesterol disturbance and a negative feedback loop between cholesterol accumulation and CYP46A1 suppression. We propose that both mitochondrial oxidative stress and negative feedback from cholesterol turnover might collectively result in the suppression of *CYP46A1* gene expression. However, we cannot exclude the possibility that enhanced ATAD3A oligomerization inhibits *CYP46A1* mRNA expression by directly interfering with its transcription. Further investigation is needed to define the mechanism of ATAD3A-mediated CYP46A1 deficiency.

Cholesterol disturbance at the MAMs of intracellular lipid rafts promotes amyloidogenic APP processing[23]. Accumulation of the APP proteolytic fragment C99 at the MAMs disrupts cholesterol

trafficking and homeostasis[62]. Our study showed that ATAD3A oligomerization is an inducer of both MAM hyperconnectivity and neuronal cholesterol accumulation. Moreover, ATAD3A immunodensity was concomitantly increased with APP in the brains of AD patients and mice. These lines of evidence raise the possibility that ATAD3A is involved in APP pathology. Indeed, our results demonstrated that a reduction in ATAD3A oligomerization by genetic knockdown or DA1 treatment normalized MAM tethering and suppressed APP processing and Aβ accumulation, resulting in reduced AD pathology. Moreover, we showed that ATAD3A oligomerization-mediated APP processing is, at least in part, dependent on CYP46A1. In this study, soluble Aβ was a driving force for aberrant ATAD3A oligomerization, as either the presence of oligomeric Aβ peptide or overexpression of APP elicited ATAD3A oligomerization. Thus, soluble Aβ-induced ATAD3A oligomerization via the ATAD3A-CYP46A1-APP signaling axis may exacerbate Aβ deposition and contribute to the AD pathology observed later in both animals and patients. Again, our observations propose a signaling pathway that supports a vicious cycle between Aβ accumulation, impaired brain lipid metabolism, and MAM hyperconnectivity.

Our recent study revealed that ATAD3A forms higher-ordered oligomers and acts as a molecular linker coupling Drp1-mediated mitochondrial fragmentation and mtDNA-mediated bioenergetic defects and triggering mitochondrial dysfunction and neuropathology in HD models[27]. The ER-mitochondria contacts also provide a platform for mtDNA replication and mitochondrial division[63]. Several mitochondrial nucleoid component proteins (e.g., twinkle) are part of a cholesterol-rich membrane structure close to the ER, suggesting that disturbed cholesterol homeostasis could affect mtDNA stability at the MAMs. ATAD3A was identified as a component of the mitochondrial nucleoid complex[64]. We previously showed that ATAD3A oligomerization suppressed mtDNA replication and mitochondrial nucleoid complex stability by disrupting TFAM-mtDNA binding[27]. Patient fibroblasts deficient in ATAD3A exhibited abnormal mtDNA distribution and replication. This phenotype was replicated by treatment with a cholesterol trafficking inhibitor[26]. Future investigation into the mechanism by which ATAD3A links cholesterol metabolism and mtDNA stability may provide a better understanding of the role of ATAD3A in neurodegenerative diseases.

Drugs that modify cholesterol homeostasis are being considered as potential therapies for AD. Cholesterol synthesis inhibitors (e.g., statins) reduce the amyloid burden in AD transgenic mouse models[65], but this positive effect awaits validation in clinical trials[66]. An inhibitor of acyl-coenzyme A: cholesterol acyltransferase also reduced amyloid pathology in an

AD mouse model[67]. These drugs modify cholesterol biosynthes is not only in the brain but also in peripheral tissues and plasma. Because cholesterol overload often happens in adult neurons[68], identifying new targets involved in neuronal cholesterol turnover may offer a new avenue for the development of AD therapeutics. We previously reported that the peptide-based ATAD3A inhibitor, DA1, had a minor effect on ATAD3A oligomerization at the steady-state and was mainly effective under stress or disease conditions[27]. In the current study, we demonstrated that DA1 significantly reduced AD-associated neuropathology, neuroinflammation, and short- and long-term cognitive deficits in AD mice. In contrast, it had no observable effects on WT animals. Notably, treatment with DA1 reduced ATAD3A oligomerization-induced CYP46A1-dependent neuronal cholesterol accumulation and had no impact on cholesterol biosynthesis, which is an advantage over cholesterol synthesis inhibitors. Therefore, DA1 or DA1-like reagents might be a potential therapeutic strategy for preventing or treating AD.

## Methods
All animal experiments in this study were conducted in accordance with protocols approved by the Institutional Animal Care and Use Committee of Case Western Reserve University and performed according to the National Institutes of Health Guide for the Care and Use of Laboratory Animals. The experiments using human postmortem tissue samples were performed with the approval of the Institutional Review Board of Case Western Reserve University.

**Reagents and antibodies**. Protein phosphatase inhibitor (P5726), protease inhibitor cocktails (P8340), voriconazole (PZ0005), and filipin (F4767) were purchased from Sigma-Aldrich. BMH (bismaleimidohexane, 22330) was purchased from Thermo Fisher Scientific. The antibody against ATAD3A (H00055210-D01, 1:1000) was from Abnova. The antibody against ATAD3A (GTX116301, 1:50) was from Genetex. The antibodies against ATPB (17247-1-AP, 1:1000), SigmaR1 (15168-1-AP, 1:1000), calnexin (17090-1-AP, 1:1000), CYP46A1 (12486-1-AP, 1:1000), and mtCO2 (55070-1-AP, 1:1000) were from ProteinTech. The antibodies against FACL4 (sc-365230, 1:1000), Tim23 (sc-514463, 1:1000), and Tom20 (sc-11415, 1:1000) were purchased from Santa Cruz Biotechnology. The antibodies against APP (ab32136, 1:5000), mtCO1 (ab14705, 1:1000), cytochrome C (ab110325, 1:10000), VDAC (ab14734, 1:2000), ClpP (ab124822, 1:2000), and synaptophysin (ab32127, 1:10000) were from Abcam. The antibodies against the C-terminal of APP (A8717, 1:5000), NeuN (A60, MAB377), FLAG (F3165, 1:2000), and β-actin (A1978, 1:10000) were obtained from Sigma-Aldrich. The Iba1 (019–19741, 1:500) antibody was from Wako Chemicals. The antibodies against GFAP (MAB360, 1:1000) and IP3R3 (AB9076,1:1000) were purchased from Millipore. The CTxB (1:200, bs-12862R) antibody was received from Bioss, and the MAP2 antibody (NB300-213, 1:500) was purchased from Novus. The PSD-95 antibody (MA1-045, 1:500) was obtained from Invitrogen. The antibody against purified anti-β-amyloid 1-16 (clone 6E10, #803001, 1:1000) was from BioLegend. The HRP-conjugated anti-mouse (31430, 1:5000) and rabbit (31360, 1:5000) secondary antibodies were from Thermo Fisher Scientific. The VeriBlot secondary antibody (HRP) (ab131366, 1:2000), which does not recognize heavy or light chains, was from Abcam. The Alexa 488 (A11034, 1:1000), 568 (A11031, 1:1000), 405 (A31553, 1:1000) fluorescent secondary antibodies were from Life Technologies and DyLight 405 (703-475-155, 1:500) was from Jackson Immuno Research. Cholesterol biosynthetic intermediates were purchased from Avanti Polar Lipids as solids: lanosterol, zymosterol, lathosterol, and desmosterol. Cholesterol-d7 standard (25,26,26,26,27,27,27-2H7-cholesterol) was purchased from Cambridge Isotope Laboratories. Efavirenz was obtained from Selleckchem (S4685). Mitotracker was purchased from Thermo Fisher Scientific (M7512).

**Computational virtual screening of ATAD3A in AD phenotypes and associated genes**
*Chemical-gene network (ChemicalGN)*. ChemicalGN contains 473,602 chemical nodes, 18,701 gene nodes, and 15,473,939 genes. We obtained genes associated with microbial metabolites from the STITCH (Search Tool for Interactions of Chemicals) database. This database contains 15,473,939 chemical-gene associations found in the human body, representing 473,602 chemicals and 18,701 human genes (data accessed in June 2019).

*Mutational phenotype-gene network (PhenGN)*. PhenGN consists of 9982 phenotype nodes, 11,021 gene nodes, and 517,381 phenotype-gene edges. We obtained a total of 517,381 systematic genetic knockouts of phenotype-gene associations (9982 phenotypes and 11,021 mapped human genes) from the Mouse Genome Database. In this study, we developed network-based prediction models leveraging these

causal phenotype-gene associations to assess how mutations in specific genes (e.g., ATAD3A) affected Alzheimer-related phenotypes.

*Pathway-Gene network (PathGN)*. PathGN contains 8868 gene nodes, 1329 pathway nodes, and 66,293 gene-pathway edges. We obtained a total of 66,293 canonical gene-pathway associations for 8868 genes and 1329 pathways from the Molecular Signatures Database (MSigDB). MSigDB is currently the most comprehensive resource for annotated pathways and gene sets.

*Protein-protein interaction network (PPIN)*. PPIN contains 22,982 gene nodes and 382,256 gene-gene edges. We obtained a total of 382,256 gene-gene associations (22,982 human genes) from BioGrid.

**Prioritization algorithm**. For the input gene ATAD3A, we prioritized other biomedical entities (genes, pathways, and phenotypes) using the context-sensitive network-based ranking algorithm that we previously developed. The output from the CSN-based algorithm was a list of phenotypes, genes, pathways, and chemicals for the input gene ATAD3A ranked based on their genetic, functional, and phenotypic relevance to the input gene.

**Evaluation of AD-associated phenotypes**. Thirteen AD-associated phenotypes were obtained from the Research Models at Alzforum database [https://www.alzforum.org/research-models/alzheimers-disease/commonly-used]), including *amyloid-beta deposits, amyloidosis, cerebral amyloid angiopathy, neurofibrillary tangles, tau protein deposits, neurodegeneration, neuron degeneration, gliosis, astrocytosis, microgliosis, abnormal synaptic transmission, abnormal long-term potentiation*, and *abnormal long-term depression*. The above network-based prioritization algorithm (input: ATAD3A; output: a list of prioritized mouse mutational phenotypes) was evaluated using these 13 AD-related phenotypes to examine how ATAD3A is phenotypically related to AD.

**Evaluation of AD-associated genes**. In this study, we used 22 genes that were strongly associated with or causally involved in AD from the Online Mendelian Inheritance in Man (OMIM) database. These 22 strong AD genes were *A2M, ABCA7, ACE, ADAM10, APBB2, APOE, APP, BLMH, HFE, MPO, MT-ND1, NOS3, PAXIP1, PLAU, PLD3, PRNP, PSEN1, PSEN2, SORL1, TF, TNF*, and *VEGFA*. The network-based prioritization algorithm (input: ATAD3A; output: a list of prioritized human genes) was evaluated using these 22 AD genes to examine how ATAD3A is genetically related to AD.

**Preparation of oligomeric Aβ$_{1-42}$**. The Aβ$_{1-42}$ (GenicBio Limited) peptides were dissolved in 1,1,1,3,3,3-hexafluoro-2-propanol (105228, Sigma-Aldrich) to a final concentration of 5 mM and placed in a chemical hood overnight. The next day, HFIP was further evaporated using a SpeedVac concentrator for 1 h. Monomer Aβ (5 mM) was prepared by dissolving Aβ peptide in anhydrous dimethyl sulfoxide (Sigma-Aldrich). The oligomeric Aβ peptides were prepared by diluting the monomer Aβ solution in Dulbecco's Modified Eagle Medium (DMEM)/F12 (1:1) (21041-025, Gibco) and then incubating at 4 °C for 24 h.

**Cell culture**. HEK293T (MilliporeSigma, 12022001), mouse hippocampal HT-22 cells (MilliporeSigma, SCC129) and Neuro2a cells (ATCC, CCL-131) were cultured in DMEM supplemented with 10% (v/v) heat-inactivated FBS and 1% (v/v) antibiotics (100 unit/mL penicillin, 100 μg/mL streptomycin). Neuro2a cells stably overexpressing human APP wildtype (APP$^{wt}$) or Swedish mutant (APP$^{swe}$, K670N and M671L APP, clone Swe.10) were obtained from Dr. Gopal Thinakaran (University of Chicago) and maintained as described above. Primary cortical neurons were isolated from E18 mouse cortex (C57ECX) from BrainBits following the manufacturer's protocol and grown according to the supplier's culturing protocol. The cells were gently resuspended in NbActiv1 culture medium (NB1, BrainBits) and plated on poly-D-lysine/laminin (P6407, Sigma-Aldrich)-coated culture plates with or without coverslips at an appropriate cell density. All cells were maintained at 37 °C and 5% CO$_2$.

**Plasmids and transfection**. Human ATAD3A-WT-Flag and ATAD3A-ΔN50-Flag plasmids were previously described[27]. Cells were transfected with plasmids using TransIT®−2020 transfection reagent (MIR5406, Mirus Bio LLC, Madison, WI), according to the manufacturer's instructions. Lenti-Syn-CYP46A1-mCherry-puro plasmid and Lenti-mCherry control plasmids were obtained from Vector-Builder Inc. Lentiviruses were generated by transfecting human embryonic kidney 293T (HEK293T) cells with plasmids encoding the envelope (pCMV-VSV-G; catalog no. 8454, Addgene), packing (psPAX2; catalog no. 12260, Addgene), and targeted open reading frame. The medium was changed 12 h after transfection, and the lentiviruses were harvested after 36 h. The lentiviruses were diluted with the corresponding medium at a 1:1 ratio, and the cells of interest were infected in the presence of Polybrene (8 μg/mL, Sigma-Aldrich) for 48 h. For knockdown of ATAD3A in Neuro2a, HT-22 cells, and mouse primary cortical neurons, cells were infected with lentivirus of control shRNA and ATAD3A shRNAs (Sigma, TRCN0000242003 and TRCN0000241479), as described previously[27].

**Generation of ATAD3A heterozygous knockout AD mice**. All mice were maintained under a 12 h/12 h light/dark cycle (light on at 6 a.m. and off at 6 p.m.) with ad libitum access to food and water under the ambient temperature at 23 °C and humidity at 40–60%. All animal experimental protocols were approved by the Institutional Animal Care and Use Committee of Case Western Reserve University. Sufficient procedures were employed to reduce the pain and discomfort of the mice during the experiments. The mice were mated, bred, and genotyped in the animal facility of Case Western Reserve University. All mice used in this study were maintained on a C57BL/6J (Strain #000664, The Jackson Laboratory) background. 5XFAD transgenic mice [Tg(APPSwFlLon,PSEN1*M146L*L286V)6799Vas, strain #034840-JAX] breeders were purchased from Jackson Laboratory.

The ATAD3A heterogeneous knockout-first mice were obtained from the Wellcome Trust Sanger Institute (Colony name: MGPY; Genetic background: C57BL/6NTac; strain #EPD0159_4_A12). A pair of loxP sites were inserted flanking ATAD3A exon 2, and a LacZ–neomycin cassette flanked with FRT was inserted in intron 1, which terminated Atad3a transcription. The knockout-first mice were then bred with Flp recombinase transgenic mice (129S4/SvJaeSor-Gt(ROSA)26Sortm1(FLP1)Dym/J, strain #003946, The Jackson Laboratory) to remove the LacZ–neomycin cassette and obtain the Atad3a-conditional knockout mice ATAD3A^flox/flox (ATAD3A^fl/fl), which contain the ATAD3A allele with exon 2 flanked by LoxP sites. ATAD3A^fl/fl mice were bred with CMV-Cre mice (B6.C-Tg(CMV-cre)1Cgn/J, strain #006054, The Jackson Laboratory) to generate CMV;ATAD3A^fl/+ heterozygous mice. 5XFAD heterozygous mice were crossed to the CMV;ATAD3A^fl/+ mice to generate 5XFAD^het;CMV;ATAD3A^fl/+ mice. Inbred, age-matched, and sex-balanced WT, CMV;ATAD3A^fl/+, 5XFAD^het;ATAD3A^+/+, 5XFAD^het;CMV;ATAD3A^fl/+ mice were used for further study.

**Systemic peptide treatment of AD mice**. Control peptide TAT and DA1 peptide (Product number P103882, Lot# 0P082714SF-01) were synthesized at Ontores (Hangzhou, China). Their purities were assessed as >90% by mass spectrometry. Lyophilized peptides were dissolved in sterile water and stored at −80 °C until use. All randomization and peptide treatments in AD mice were prepared by an experimenter not associated with the behavioral and neuropathology analysis. 5XFAD transgenic mice and their age-matched and sex-balanced WT littermates were implanted with a 28-day osmotic pump (Alzet, Cupertino CA, Model 2004) containing either TAT control peptide or DA1 peptide, which delivered the peptides at a rate of 1 mg/kg/day, from age of 1.5 to 9 months. The pump was replaced once every 4 weeks.

**Behavioral analysis**. All behavioral analyses were conducted by an experimenter who was blinded to the genotypes and treatment groups. All mice were subjected to a series of behavioral measurements to monitor locomotor activity (open field test), spontaneous spatial working memory (Y-maze test), long-term spatial learning, and memory functions (Barnes maze test). Body weights and survival rates were recorded throughout the study period.

*Y-maze test*. On the test day, mice (6 months old) were brought to the testing room one hour before performing the Y-maze test to allow habituation. The mice were placed in the middle of the Y-maze and allowed to explore the three arms for 6 min. During exploration, the arm entries were recorded. The equipment was cleaned after every test to avoid odor disturbance. Spontaneous alternation was defined as a successive entry into three different arms on overlapping triplet sets.

*Barnes maze test*. On the test day, the mice were brought to the testing room 30 min before performing the Barnes maze test to allow habituation. Briefly, all the testing mice received three consecutive days of trials, with three trials each day. After being placed in the center of the platform at the beginning of each trial, the mice were allowed to explore for 3 min to find the target escape box. Mice that failed to enter the target escape hole in the given time were led to it by the operator. Mice were allowed to remain in the target hole for 2 min before returning to the home cage. After completing the 3-day trials, the mice were examined on days 5 and 12 with one test to monitor the long-term spatial learning and memory activities. The maze and the escape box were cleaned carefully after each trial to avoid odor disturbance. All the trials and tests were recorded with a video system. The total time to enter the target escape box (latency to the target box) and the number of times the wrong holes were explored (the total errors) were recorded.

*Nest building performance test*. Mice (8.5 months old) were subjected to the nest building test. Briefly, ~1 h before the dark phase (light on/off 6:00 a.m./6:00 p.m.), a mouse was transferred to a new cage containing a new nestlet (around three grams) made from pressed cotton. The performance of nest building by the mouse was evaluated the next morning using a rating scale of 1–5. The rating score was assessed: 1, >90% intact, not noticeably touched; 2, 50–90% intact, partially torn; 3, 10–50% intact, mostly shredded nestlet without an identifiable nest site; 4, 0–10% intact, an identifiable but flat nest; 5, 0–10% intact, crater-shaped nest. The untorn nestlet pieces in each cage were weighed, and the percentage of usage of the nestlet was obtained by subtracting the remaining nestlet weight from the initial nestlet weight.

*Open field test*. The locomotor activity of all experimental mice was assessed in an open field at six months of age. Briefly, the mice were placed in the center of an activity chamber (Omnitech Electronics) and allowed to explore the chamber while being tracked using an automated infrared tracking system (Vertax, Omnitech Electronics). A 12-h locomotor activity analysis was performed.

**Human postmortem brain samples**. All postmortem brain samples were collected by the National Institutes of Health (NIH) NeuroBioBank (NBB; https://neurobiobank.nih.gov/) under the approval of the Institutional Review Boards (IRB) and the institution's Research Ethics Board. All brains were donated to the NBB by informed consent through the Brain and Tissue Repositories sites. Donation was voluntary and had no financial benefits. All brain specimens donated to the NIH NBB were assessed and reviewed by board-certified neuropathologists. A standard assessment was performed to document possible neuropathologies and establish a disease condition diagnosis. In addition, postmortem blood was sampled and submitted for serology and toxicology testing. The human postmortem brain samples used in the experiments were obtained from the NBB under a material transfer agreement between the NIH and Case Western Reserve University. The experiments were performed with the approval of the IRB of Case Western Reserve University. The detailed information for the human postmortem brain samples is listed in Supplementary Fig. 2b.

**Mouse brain mitochondrial sub-compartmental fractionation**. Isolation of mitochondrial sub-compartmental fractions, including the ER, mitochondria, and ER-mitochondrial contact sites, was performed as previously described[69]. Briefly, 3-month-old WT mice (C57BL/6) were deeply anesthetized and transcardially perfused with PBS. The brain tissues were washed and homogenized with ice-cold mitochondrial isolation buffer (225 mM mannitol, 75 mM sucrose, 0.5% BSA, 0.5 mM EGTA, and 30 mM Tris–HCl, pH 7.4) on ice. Cell debris and nuclei were removed by centrifugation, and the supernatants were subjected to further differential centrifugation to get the crude mitochondrial fraction in the pellet. The supernatants were further centrifuged at $100{,}000 \times g$ for 1 h to obtain the ER fraction. The crude mitochondrial fraction was resuspended in mitochondrial resuspending buffer (MRB, 250 mM mannitol, 0.5 mM EGTA, and 5 mM HEPES, pH 7.4) to the final volume of 2 mL, and the crude mitochondrial suspension was layered on the top of the Percoll medium (225 mM mannitol, 25 mM HEPES pH 7.4, 1 mM EGTA and 30% Percoll (vol/vol)). The pure mitochondrial and MAM fractions were isolated by centrifugation at $95{,}000 \times g$ for 30 min, washed to remove the Percoll, and further purified by centrifugation to eliminate contaminants. All fractions were reconstituted in mitochondrial buffer and stored at −80 °C until analysis.

**In situ proximity ligation assay**. PLA was performed using the Duolink® In Situ Red Starter Kit (mouse/rabbit, DUO92101, Sigma). Briefly, fixed cells or brain sections were permeabilized and blocked with PLA blocking buffer for 1 h at 37 °C and incubated with the indicated primary antibodies overnight at 4 °C. The samples were then incubated with the PLA probes (Anti-Rabbit PLUS and Anti-Mouse MINUS) for 1 h at 37 °C, followed by the ligation and amplification steps. The PLA signal was visible as a distinct fluorescent spot and analyzed by confocal microscopy (Fluoview FV1000, Olympus). The number of fluorescent signals was quantitated using NIH ImageJ software.

**ATPase activity measurement**. Cells and mouse brain tissues were harvested and lysed in total lysis buffer. The supernatants were incubated with ATAD3A or Flag antibodies overnight at 4 °C, followed by incubation with protein A/G beads (Santa Cruz Biotechnology, sc-2003,) for 2 h at 4 °C. The immunoprecipitates were washed with lysis buffer and then ATPase activity was measured using a commercially available kit (Abcam, ab234055).

**Label-free proteomics**. Each frozen mouse cortex ($n = 3$ mice per group) was collected in a 1.5-mL tube containing 300 μL of 2% SDS and protease inhibitor cocktail. The samples were incubated on ice for 30 min and then sonicated with a probe sonicator at 50% amplitude, followed by vortexing. This cycle was repeated four times, with samples placed on ice between each round. Following lysis, the samples were processed using a filter-aided sample preparation cleanup protocol using Amicon Ultra MWCO 3K filters (Millipore, Billerica, MA). The samples were reduced and alkylated on the filters with 10 mM dithiothreitol (Acros, Fair Lawn, NJ) and 25 mM iodoacetamide (Acros, Fair Lawn, NJ), respectively, and then concentrated to a final volume of 40 μL in 8 M urea. Protein concentration was measured using the Bradford method, according to the manufacturer's instructions (Bio-Rad, Hercules, CA).

Following reduction and alkylation, total protein (10 mg) was subjected to enzymatic digestion. The urea concentration was adjusted to 4 M using 50 mM Tris (pH 8), and then proteins were digested with mass spectrometry-grade lysyl endopeptidase (Wako Chemicals, Richmond, VA) for 2 h at 37 °C using an enzyme to substrate ratio of 1:40. The urea concentration was further adjusted to 2 M using 50 mM Tris (pH 8), and the lysyl peptides were digested with sequencing-grade trypsin (Promega, Madison, WI) at 37 °C overnight using an enzyme to substrate

ratio of 1:40. Finally, the samples were diluted in 0.1% formic acid (Thermo Scientific, Rockford, IL) before LC-MS/MS analysis.

The peptide digests (320 mg, 8 µL) were loaded onto a column with blanks in between for a total of four LC/MS/MS runs. The resulting data were acquired using an Orbitrap Velos Elite mass spectrometer (Thermo Electron, San Jose, CA) equipped with a Waters nanoACQUITY LC system (Waters, Taunton, MA). The peptides were desalted on a trap column (180 µm × 20 mm, packed with C18 Symmetry, 5 µm, 100 Å, Waters, Taunton, MA) and resolved on a reversed-phase column (75 µm × 250 mm nano column, packed with C18 BEH130, 1.7 µm, 130 Å) (Waters, Taunton, MA). Liquid chromatography was carried out at an ambient temperature at a flow rate of 300 nL/min using a gradient mixture of 0.1% formic acid in water (solvent A) and 0.1% formic acid in acetonitrile (solvent B). The gradient ranged from 4 to 44% solvent B over 210 min. The peptides eluting from the capillary tip were introduced into the nanospray mode with a capillary voltage of 2.4 kV. A full scan was obtained for the eluted peptides in the range of 380–1800 atomic mass units, followed by 25 data-dependent MS/MS scans. The MS/MS spectra were generated by collision-induced dissociation of the peptide ions at a normalized collision energy of 35% to generate a series of b- and y-ions as major fragments. In addition, a one-hour wash was included between each sample. The proteins were identified and quantified using PEAKS 8.5 (Bioinformatics Solutions Inc., Waterloo, ON, CA).

**Total RNA isolation and real-time quantitative RT-PCR**. Total RNA was isolated using the RNeasy Mini Kit (74004, QIAGEN) or TRIzol Reagent (15596-026, Invitrogen), and cDNA was synthesized from 0.5–1 µg of total RNA using the QuantiTect Reverse Transcription Kit (205311, QIAGEN). qRT-PCR was performed with QuantiTect SYBR Green (204143, QIAGEN) and analyzed using the StepOnePlus Real-Time PCR System (Thermo Fisher Scientific). Three replicates were performed with each biological sample, and the expression values of each replicate were normalized against GAPDH cDNA using the $2^{-\Delta\Delta CT}$ method. The primers used in this study are presented in Supplementary Table 1.

**LDH assays**. Cell death was determined using the Cytotoxicity Detection Kit (LDH), according to the manufacturer's protocol (Roche, REF 001,11,644,793).

**GC-MS measurements of sterol profiles**. Brain samples were resuspended in a 2:1 chloroform/methanol mixture and homogenized. Cholesterol-d7 standard (25,26, 26,26,27,27,27-2H7-cholesterol, Cambridge Isotope Laboratories) was added before drying under nitrogen stream and derivatization with bis(-trimethylsilyl)trifluoroacetamide/trimethylchlorosilane to form trimethylsilyl derivatives. Following derivatization samples were analyzed by gas chromatography/mass spectrometry using an Agilent 5973 Network Mass Selective Detector equipped with a 6890 gas chromatograph system and a HP-5MS capillary column (60 m × 0.25 mm × 0.25 µm). The samples were injected in splitless mode and analyzed using electron impact ionization. Ion fragment peaks were integrated to calculate sterol abundance, and quantitation was performed relative to cholesterol-d7. The following m/z ion fragments were used to quantitate each metabolite: cholesterol-d7 (465), zymosterol (456), desmosterol (456, 343), lanosterol (393), and lathosterol (458). Calibration curves were generated by injecting varying concentrations of sterol standards and maintaining a fixed amount of cholesterol-D7.

**Total cholesterol and 24-OHC measurements by ELISA**. The total cholesterol content was measured using the total cholesterol assay kit (STA-384, Cell Biolabs), according to the manufacturer's instructions. Briefly, the total lipids were extracted from cells or cortex brain tissue samples using a mixture of chloroform, isopropanol, and NP-40 (7:11:0.1). The homogenates were centrifuged at $15,000 \times g$ for 10 min, and the supernatants were collected and dried to remove the organic solvent. The dried lipids were dissolved in 1× assay diluent for further quantification assay by adding cholesterol reaction reagent (cholesterol oxidase 1:50, HRP 1:50, colorimetric probe 1:50 and cholesterol esterase 1:250 in 1× assay diluent). The calculated amount of total cholesterol for each sample was normalized to the total cell number or the weight of the cortex tissue.

The 24-OHC levels in the mouse serum or cell culture samples were determined using the mouse 24-OHC ELISA kit (MBS7256268, MyBioSource), following the manufacturer's protocol. The 24-OHC concentration was extrapolated from a standard curve and normalized to the total cell number.

**Immunofluorescence**. Cells were grown on coverslips, fixed with 4% paraformaldehyde for 20 min at room temperature, permeabilized with 0.1% Triton X-100 in PBS, and blocked with 2% normal goat serum. The cells were incubated with the indicated primary antibodies overnight at 4 °C. After washing with PBS, the cells were incubated with Alexa Fluor 488/568 or 405/568 secondary antibody (1:500; Thermo Fisher Scientific) for 2 h at room temperature. The nuclei were counterstained with DAPI (1:10,000; Sigma-Aldrich). Images of the staining were acquired using a Fluoview FV 1000 confocal microscope (Olympus).

For immunofluorescence staining of mouse brain sections, mice were deeply anesthetized and transcardially perfused with 4% paraformaldehyde in PBS. Brain sections were permeabilized with 0.2% Triton X-100 in TBS-T buffer, followed by

blocking with 5% normal goat serum. The brain sections were incubated with the indicated primary antibodies overnight at 4 °C and then stained with secondary antibodies. Images of the staining were acquired using a Fluoview FV 1000 confocal microscope (Olympus). For detection of cholesterol, brain sections were stained with filipin (F4767, Sigma-Aldrich) at room temperature in the dark. Filipin staining was imaged using an all-in-one fluorescence microscope (Keyence BZ-X710).

All quantification of immunostaining was performed using ImageJ software. The same image exposure times and threshold settings were used for all sections from all the experimental groups. Quantitation was performed blinded to the experimental groups.

**Immunohistochemistry**. Paraffin-embedded brain sections (10 µm, coronal) were staining for ATAD3A (ab112572, Abcam; or GTX116301, GeneTex) using the IHC Select HRP/DAB kit (Millipore). Quantification of the ATAD3A immunostaining was conducted using NIH ImageJ software. The same image exposure times and threshold settings were used for all sections from all treatment groups. Quantitation was performed blinded to the experimental groups.

**Fluoro-Jade C Staining**. Fluoro-Jade® C Staining was performed following the manufacturer's protocol (Biosensis Ready-to-Dilute (RTD)™ Fluoro-Jade® C Staining Kit, TR-100-FJT, Biosensis). Briefly, brain sections were incubated with NeuN antibody diluted in PBS containing 0.3% Triton-100 and 3% goat serum overnight at 4 °C and then stained with Alexa Fluor 568 secondary antibody for 2 h at room temperature. After washing in PBS, the brain sections were then incubated with 0.06% potassium permanganate solution for 5 min to block the background fluorescence and optimize the signal contrast. Following washing with distilled water, the sections were stained with Fluoro-Jade C/DAPI solution for 10 min in the dark. The slides were then rinsed thoroughly with distilled water and dried at 50–60 °C for 5 min in the dark. The dried slides were cleared using xylene and then permanently mounted. FJC and NeuN double-labeled degenerating neurons were visualized using a Fluoview FV1000 confocal microscope (Olympus).

**Synaptic density measurement in primary cortical neurons**. The synaptic density of primary cortical neurons was measured by counting the PSD95+ synaptophysin+ clusters adhered to the dendrites. Briefly, the primary neurons were fixed with 4% paraformaldehyde for 30 min, followed by permeabilization and blocking at room temperature. The cells were then incubated with primary antibodies against PSD95 (MA1-045, Invitrogen) and synaptophysin (ab32127, Abcam) overnight at 4 °C, followed by Alexa 488- and Alexa 568-labeled secondary antibodies, respectively. The number of synapses per micron of dendrites was calculated[70].

**Golgi staining and quantification of dendrite spine density**. Golgi-cox staining was performed using the NovaUltra Golgi-Cox Stain Kit (IHCWorld, SKU IW-3023). Briefly, mice were deeply anesthetized and transcardially perfused with PBS. The mouse brains were immersed in the Golgi-Cox Solution in the dark at room temperature. After 2 days of immersion, fresh Golgi-Cox Solution was added to the samples, which were incubated at room temperature in the dark for an additional 14 days, according to the manufacturer's instructions. After washing with PBS for two days, serial coronal sections (200-µm thick) were cut with a Vibratome Series 1000 Sectioning System. The coronal sections were washed with water and stained with Post-Impregnation Solution for 10 min in the dark at room temperature. Following three washes with water, the brain sections were mounted on Superfrost Plus slides (Thermo Scientific). The dendritic spine images were acquired using a 100× oil objective. Cortical pyramidal neurons were selected for analysis. The dendritic spines were counted in 50–100 µm segments that were at least 50 µm away from the cell body. The total spine density was measured using the NIH Image J plug-in simple neurite tracer.

**Co-immunoprecipitation**. Tissues were lysed in total cell lysate buffer (50 mM Tris-HCl [pH 7.5], 150 mM NaCl, 1% Triton X-100, and protease inhibitor cocktail). Total lysates were incubated with the indicated antibodies overnight at 4 °C, followed by the addition of protein A/G beads for 2 h at room temperature. The immunoprecipitates were washed four times with cell lysate buffer and analyzed by western blotting.

**Western blotting**. The protein concentration in each sample was determined using protein assay dye reagents (Bio-Rad). The proteins were resuspended in Laemmli buffer, separated using sodium dodecyl sulfate-polyacrylamide gels, and transferred to nitrocellulose membranes. The membranes were probed with the indicated antibodies, and the specific proteins were visualized by electrochemiluminescence. The chemiluminescence signals were captured by X-ray films or cSeries Capture Software 1.9.8.0403 (C600 azure biosystem).

**Quantification and statistical analysis**. Sample sizes were determined by power analysis based on pilot data collected in our laboratory or published studies. For the animal studies, we used 13–33 mice/group for the behavioral tests, $n = 3$–8 mice/group

for the biochemical analyses, $n = 3–11$ mice/group for the pathology studies, and $n = 4–13$ mice/group for the cholesterol metabolic pathway analyses. Both male and female mice were used through all the study, mix-sex analysis was used in all the experiments. For the cell culture studies, we performed each experiment with at least three independent replicates. For the animal studies, we ensured randomization and blinded evaluation. All imaging analyses were conducted by an observer blinded to the experimental groups. No samples or animals were excluded from the analysis.

The data were analyzed using GraphPad Prism 9.0 software. The unpaired Student's $t$ test (two-tailed) was used for comparisons between two groups. Comparisons between three or more independent groups were performed using one-way analysis of variance (ANOVA), followed by Tukey's multiple comparison's test. Comparisons of the effect of independent variables on a response variable were performed using two-way ANOVA. The data are presented as the mean ± standard errors of the mean. Statistical parameters are presented in each figure legend. Values of $p < 0.05$ were considered statistically significant.

**Reporting summary**. Further information on research design is available in the Nature Research Reporting Summary linked to this article.

## Data availability

Data supporting the findings of this study are provided within the paper and its supplementary information. Source data are provided with this paper and all statistical data are presented in Source Data file. The mass spectrometry proteomics data have been deposited to the ProteomeXchange Consortium via the PRIDE partner repository with the dataset identifier PXD031523 (http://www.ebi.ac.uk/pride/archive/projects/PXD031523). Source data are provided with this paper.

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

## Acknowledgements

This study was supported by grants from the US National Institutes of Health (R01AG065240, R01NS115903, and R21NS107897 to X.Q.; R01AG057557, R01AG061388 and R56AG062272 to R.X.; R01NS115867 to D.J.A.; F30MH116581 and TL1 TR000441 to Z.H., and Dr. Ralph and Marian Falk Medical Research Trust-Transformative Award and Harrington Rare Disease Scholar Award to X.Q and D.J.A., and The Mathers Foundation and Thomas F. Peterson, Jr. to D.J.A. The authors thank the NIH NeuroBioBank for providing the human postmortem brain tissues and brain donors and their families for the tissue samples used in this study. The study with the mouse genotypes was supported by a P30 core grant for vision research (NIH 5P30EY011373). The authors thank I. Bederman and M. Drumm for technical support.

## Author contributions

Y.Y.Z., D.H., and R.H.W. performed cell culture experiments and biochemical analyses of animal models and patient samples, conducted blinded animal behavioral analysis, performed in vivo analysis on AD-associated pathology, and drafted the manuscript. X.Y.S. bred the heterozygous knockout ATAD3A mice, 5XFAD mice, and the triple-crossed mice. P.R. performed high resolution imaging; Z.H. and D.A. carried out the GC-MS study. K.L. performed the proteomics analysis, and Q.W. and R.X. performed the computational analysis of ATAD3A in the AD human database. X.Q. conceived, designed, and supervised all the studies and edited the manuscript.

## Competing interests

The authors declare no competing interests.
