## [Peer Review File · Nature Communications]

Reviewers' Comments:

Reviewer #1:

Remarks to the Author:

This is an interesting report from Zao et al., demonstrating the role of ATAD3A oligomerization in promoting neuropathology and cognitive deficits in AD 5xFAD mouse model. Using ATAD3 knockdown mice and DA1 peptide, they found reduction in cerebral A β , amyloid pathology, reduced oligomer form of ATAD3A, balance of cholesterol clearance, and protection of synaptic loss and cognitive dysfunction in 5xFAD mice. They also applied in vitro neuronal cell line HT22 and N2a-APP cells to test the effect of ATAD3 on oligomer A β -induced cell injury and oligomerization of ATAD3A. Furthermore, they found correlation of ATAD3A oligomer forms with CYP46A1 expression. It appears that ATAD3A regulates cholesterol metabolism and APP processing through CYP46A1 or cooperation with CYP46A1 gene. The studies are well outlined and organized using different model systems. Although the similar effect of oligomer form of ATAD3A on HD as author described previously, the impact of ATAD3A on amyloid pathology remains to be determined. Thus the significance of these studies is high to enhance our understanding of amyloid pathology related to ATAD3A dysregulation. There are some concerns needed to be addressed with the additional experiments.

1. To determine the specific effect of oligomer form of ATAD3A on A β -induced APP process and neuronal functional or cholesterol in an in vitro or in vivo, it is essential to test the direct effects of oligomer ATAD3A excluding monomer of ATAD3A (not full length of protein). Either knockdown or knockout of gene will suppress entire gene and protein expression including monomers and oligomer form of ATAD3A protein.
2. It is unclear the amount of oligomer ATAD3 on MAM and mitochondrial fraction in human AD brain and AD mice. This requires the quantification of immunoblotting of cellular fractions from AD-affected and spared regions of human AD patient and AD mouse brains in addition to immunostaining. Are there differences of oligomerization of ATAD3 between AD-affected regions and spared regions?
3. Lack of data showing effect of oligomer A β on oligomer formation of ATAD3A on primary cortical neurons, though they showed results generated from HT-22 and N2a cells, which properties are different from primary cortical neurons such as their response to A β toxicity etc.
4. It is unclear whether ATAD3A physically localizes in mitochondrial matrix only and translocate to MAM under pathological condition. Comparison of distribution of ATAD3A protein (full length and oligomers) on mitochondrial matrix with MAM in physiological and pathological condition is important to dissect its role. For example, Figure S2A showed a mouse brain mitochondrial sub-compartmental fractions, however, there are no data showing the distribution of ATAD3 from AD mouse or DA1-treated mouse brain compared to WT mouse. Similarly, the distribution of ATAD3A on mitochondrial sub-compartmental fraction from A β treatment compared to control group is not clear.
5. What is the sequence for TAT and DA1 peptide? It will be interesting to see the effect of the sequences from the first 50 amino acids as a control peptide to compare a truncated ATAD3A mutant.
6. It is clear the significant association of ATAD3 with CYP46A1 on APP process and cholesterol metabolism. However, the additional experiments are required to support that ATAD3 regulates amyloid pathology dependent on CYP46A1 in vivo, such as double transgenic ATAD3/CYP46A1 mice.
7. Rationale for using heterozygous or homozygous 5xFAD mice needs to be clarified.

Reviewer #2:

Remarks to the Author:

In this manuscript, Zhao et al identify the oligomerization of ATAD3A, a mitochondria-associated AAA ATPase, as an Alzheimer's disease (AD) associated molecular anomaly that is induced by amyloid-beta and promotes the amyloidogenic pathway in vitro and in vivo. ATAD3A was shown to oligomerize and concentrate at the mitochondria-associated ER membranes (MAMs), where it represses the transcription of CYP46A1, an enzyme that mediates cholesterol hydroxylation and thus its catabolism. Consequently, cholesterol accumulates in AD models as a result of ATAD3A oligomerization and this accumulation promotes APP processing. Importantly, reducing ATAD3A

oligomerization genetically in a haploinsufficient ATAD3A background or pharmacologically with a brain penetrant inhibitory peptide restores normal levels of CYP46A1 in CNS neurons from the 5XFAD mouse model, corrects cholesterol metabolism. and decreases amyloid burden, gliosis and cognitive deficits.

This is a thorough study on the pathogenic role of ATAD3A oligomerization in AD models that involves MAM dyshomeostasis, impact on cholesterol metabolism and APP processing. There are a lot of in vitro and in vivo data that support the hypothesis that CYP46A1 downregulation may account for these downstream phenotypes and a rather compelling case that downregulating ATAD3A genetically or preventing its oligomerization with a peptide confers benefits in the 5XFAD model. The missing links are: what mechanism causes ATAD3A oligomerization? What are the relevant "clients" of this AAA ATPase in the MAMs? How does ATAD3A oligomerization downregulate CYP46A1 transcription? Given the strength and large amount of data provided, these questions may be best suited for follow-up studies. However, there are a few aspects that need to be strengthened to make this manuscript suitable for publication in NCOMMs in my view.

1. The authors use the proximity ligation assay extensively to examine MAM proteins and results are overall rather compelling. The downside is that there is no information on ER or mitochondria morphology. Does ATAD3A oligomerization induced by amyloid beta affect mitochondria fusion/fission? Can they show greater colocalization between ER and mitochondria with the appropriate ER and mitochondria markers?
2. Does ATAD3A oligomerization increase its AAA ATPase activity? In other words, is it a gain of function that results in its pathogenic effect? What is the impact of an ATPase-dead mutant on MAMs?
3. The DA1 peptide was used before in another in vivo study. Can the authors provide drug levels in the CNS of the 5XFAD mice? It's always difficult to interpret efficacy data without knowing anything about drug levels.
4. Does Efavirenz, a CYP46A1 agonist, correct the phenotypes induced by ATAD3A oligomerization in vitro?

Minor comments

5. Page 3, 2nd paragraph: "Moreover, the amyloid precursor protein (APP) and APP processing γ -secretases, presenilin-1 (PS1) and presenilin-2 (PS2), are highly enriched in the MAMs relative to other cell compartments, such as the plasma membrane, mitochondria, and ER". I don't think that APP is enriched in MAMs compared to other compartments. It is also abundant in the TGN, plasma membrane and endosomal compartment so perhaps a more conservative view is that APP is "present in the MAMs". Presenilins, on the other hand, may well be enriched in the MAMs.

6. Can the authors provide western blot data showing that the 5XFAD brains have lower CYP46A1 protein levels, consistent with the proteomics?

Reviewer #3:

Remarks to the Author:

In this study, the authors showed ATAD3A oligomerization by amyloid beta is critical in AD pathogenesis by regulating cholesterol turnover and MAM integrity, thereby exacerbating AD pathology. Although genetic ATAD3A deficiency and pharmacological inhibition to inhibit ATAD3A oligomers reduced AD pathology in AD model mice, there were some problems that need to be addressed.

Major points:

1. The authors discovered that the presence of amyloid beta (in vitro, AD patients, and 5XFAD mice) induces the oligomerization of ATAD3A and the levels of ATAD3A monomer does not change when using beta-mercaptoethanol to limit the oligomerization. However, the immunostaining

signals of ATAD3A in AD patients and AD model mice are markedly different compared to controls. If so, is this antibody recognizing oligomers of ATAD3A? If the antibody against ATAD3A used by the authors could detect the entire ATAD3A form, including monomers and oligomers, then the staining pattern of ATAD3A would be expected to be similar in both control and AD conditions. Therefore, it seems likely that the antibody will need to be validated using other antibodies whether it correctly recognizes ATAD3A.

In addition, the staining pattern of ATAD3A is distributed throughout the cytoplasm, not in the form of mitochondria or MAMs. And in the AD context, it appears that ATAD3A expression was induced rather than the oligomerization of ATAD3A.

2. In Figure 2E, the integrity of MAM appears to be affected only by the knockdown of ATAD3A regardless of the presence of amyloid beta. It appears that the function of ATAD3A itself, not oligomerization, has a significant impact on MAM integrity. There appears to be insufficient evidence for the effect of ATAD3A oligomerization on MAM integrity.

3. The authors used a mutant form of ATAD3A (delta N50) to promote the oligomerization of ATAD3A. In results where authors used the mutant, the effect of N50 does not show a significant effect compared to WT (no statistical significance), but rather it seems to be an overexpression effect of ATAD3A. Again, it is difficult to say that this is a phenomenon caused by oligomerization.

4. Representative staining images are missing in Figures 3E, 4B, and 5H. If the space for results is insufficient, it also should be added in the supplementary figure.

5. It seems necessary to confirm the staining pattern of Jade C in Figure 4G. As Jade C-positive signals are known to be mainly present on the cell body, it is difficult to find a co-localized pattern confined to DAPI in the result. Authors need to determine whether the staining results are indicative of actual degenerating neurons.

6. Figure 5B shows staining of CYP46A1 in the hippocampal CA1 region. In this regard, Figure 1 did not show the staining results of ATAD3A in the hippocampal CA1, but it would be nice if it showed pattern changes of ATAD3A in the hippocampus.

Minor points:

1. The authors used the phrase 'ATAD3A activation' in the middle of the paper, so the concepts of 'oligomerization' and 'activation' seem confusing. It should be unified through 'oligomerization'.

2. The authors need to modify the color index of the heatmap in Figure 5A. In general, red indicates an increase and blue indicates a decrease. Also, the authors need to write the value of the color index.

Response letter

RE: ATAD3A oligomerization promotes neuropathology and cognitive deficits in Alzheimer's disease models (manuscript reference NCOMMS-21-14370)

We thank the reviewers for their constructive comments and criticisms. In the following sections, we provide point-by-point responses to these comments. Because of the additional experiments and modifications requested, we have made a number of changes to the revised manuscript. These are described in detail below. Figures, results, figure legends, discussions, and methods are added at the appropriate places in the revised manuscript. All newly added sentences are underlined and marked in blue in the manuscript. We believe that these changes have significantly improved our manuscript and we hope that the editor and reviewers will now find the revised manuscript suitable for publication.

REVIEWER COMMENTS

Reviewer #1 (Remarks to the Author):

This is an interesting report from Zhao et al., demonstrating the role of ATAD3A oligomerization in promoting neuropathology and cognitive deficits in AD 5xFAD mouse model. Using ATAD3 knockdown mice and DA1 peptide, they found reduction in cerebral A β , amyloid pathology, reduced oligomer form of ATAD3A, balance of cholesterol clearance, and protection of synaptic loss and cognitive dysfunction in 5xFAD mice. They also applied in vitro neuronal cell line HT22 and N2a-APP cells to test the effect of ATAD3 on oligomer A β -induced cell injury and oligomerization of ATAD3A. Furthermore, they found correlation of ATAD3A oligomer forms with CYP46A1 expression. It appears that ATAD3A regulates cholesterol metabolism and APP processing through CYP46A1 or cooperation with CYP46A1 gene. The studies are well outlined and organized using different model systems. Although the similar effect of oligomer form of ATAD3A on HD as author described previously, the impact of ATAD3A on amyloid pathology remains to be determined. Thus the significance of these studies is high to enhance our understanding of amyloid pathology related to ATAD3A dysregulation. There are some concerns needed to be addressed with the additional experiments.

1. To determine the specific effect of oligomer form of ATAD3A on A β -induced APP process and neuronal functional or cholesterol in an in vitro or in vivo, it is essential to test the direct effects of oligomer ATAD3A excluding monomer of ATAD3A (not full length of protein). Either knockdown or knockout of gene will suppress entire gene and protein expression including monomers and -oligomer form of ATAD3A protein.

Response: We appreciate the reviewer's comment. Below, we have summarized the data presented in our original manuscript supporting a direct impact of ATAD3A oligomers on cellular phenotypes of AD *in vitro* and *in vivo*. Specifically,

- 1) Treatment with DA1, a selective inhibitor of ATAD3A oligomerization, abolished ATAD3A oligomerization, and reduced APP processing, cholesterol accumulation and synaptic loss in toxic A β -exposed neuronal cells and the 5XFAD AD mouse model. Significantly, suppression of ATAD3A oligomerization by treatment with DA1 reduced AD-associated neuropathology and cognitive deficits in 5XFAD AD mice (Fig. 4; 5d-e, 5g-i, 5k; 6b,d,e; 7c, d, f, g; Supplementary Fig. 4, 5c,d,g, j; 6c in the original manuscript).
- 2) Expression of ATADA- Δ N50-Flag variant which enhances ATAD3A oligomerization induced cholesterol accumulation, APP processing and synaptic loss in neuronal cells compared with ATAD3A-WT-Flag-expressing cells (Fig. 5l, m; 6c; 7a in the original manuscript)

These findings demonstrate that ATAD3A oligomerization is one of the critical factors eliciting AD-associated pathology.

We previously reported that deletion of the first 50 amino acids of ATAD3A (ATAD3A- Δ N50) enhanced ATAD3A oligomerization, whereas ATAD3A- Δ CC, in which the two coiled-coil (CC) domains in the N-terminus

of ATAD3A are deleted, abolished ATAD3A oligomer formation (Zhao *et al.*, *Nature Communications*, 2019). To further address the reviewer's comment, we transfected either ATAD3A- Δ N50-Flag or ATAD3A- Δ CC-Flag variant in Neuro2a cells and treated them with DA1 or the control peptide TAT. In these cell cultures, we then compared APP processing and cholesterol content. We found that overexpression of ATAD3A-WT or ATAD3A- Δ N50 (oligomerization prone) significantly elevated the cholesterol level and the cleaved product of APP processing (C99). In contrast, suppression of ATAD3A oligomerization by DA1 treatment abolished these aberrant events induced by ATAD3A-WT or - Δ N50 variant. In parallel, overexpression of ATAD3A- Δ CC-Flag variant showed comparable effects on these events to those in control vector-expressing cells (Fig. 6c, Supplementary Fig. 6j in the revised manuscript). These findings further demonstrate that ATAD3A oligomers directly induce APP processing and cholesterol accumulation.

The following sentences were added to the Results section (page 14-15).

“Overexpression of ATAD3A-WT-Flag and ATAD3A- Δ N50-Flag variants in Neuro2a cells increased the total cholesterol content and decreased the 24-OHC content compared to cells expressing the control vector, with a worse extent observed in ATAD3A- Δ N50-Flag-expressing cells (Fig. 5l, m, Supplementary Fig. 6j). **In contrast, DA1 treatment abolished ATAD3A-WT- or - Δ N50-induced cholesterol accumulation, and overexpression of ATAD3A- Δ CC-Flag variant, which blocks ATAD3A oligomer formation, showed a comparable effect on cholesterol content to that seen in control vector-expressing cells (Supplementary Fig. 6j). These data support a direct effect of ATAD3A oligomerization on cholesterol accumulation.”**

“In contrast, blockage of ATAD3A oligomerization by either expression of the ATAD3A- Δ CC-Flag variant or treatment with DA1 abolished the C99 production (Fig. 6c), indicative of a direct impact.” (page 16)

2. It is unclear the amount of oligomer ATAD3 on MAM and mitochondrial fraction in human AD brain and AD mice. This requires the quantification of immunoblotting of cellular fractions from AD-affected and spared regions of human AD patient and AD mouse brains in addition to immunostaining. Are there differences of oligomerization of ATAD3 between AD-affected regions and spared regions?

Response: Per the reviewer's suggestion, we harvested tissue from different brain regions of three-month-old wildtype (WT) and 5XFAD mice, including the cortex, hippocampus, thalamus, midbrain, striatum, and cerebellum. We then determined the levels of ATAD3A oligomers. ATAD3A oligomers were increased in the cortex, hippocampus, and thalamus of three-month-old 5XFAD mice relative to age-matched WT mice. However, the levels of ATAD3A oligomers in the midbrain, striatum, and cerebellum were comparable in WT and 5XFAD mice. These findings are consistent with the region-specific A β aggregations and human APP expression in the early stage of AD. We have added the following text to the Results section of the revised manuscript.

“ATAD3A oligomers were also elevated in the cortex, hippocampus and thalamus of 5XFAD AD mice, but not in other brain regions (Fig. 1e, Supplementary Fig. 2c), consistent with region-specific A β aggregation and human APP expression³³.” (page 6)

3. Lack of data showing effect of oligomer A β on oligomer formation of ATAD3A on primary cortical neurons, though they showed results generated from HT-22 and N2a cells, which properties are different from primary cortical neurons such as their response to A β toxicity etc.

Response: In response to the reviewer's comment, we examined the level of ATAD3A oligomers in primary cortical neurons isolated from the E18 mouse cortex after treatment with toxic A β oligomers. The results consistently showed that toxic oligomeric A β induced the formation of ATAD3A oligomers in mouse primary cortical neurons (Fig. 1b in the revised manuscript).

4. It is unclear whether ATAD3A physically localizes in mitochondrial matrix only and translocate to MAM under pathological condition. Comparison of distribution of ATAD3A protein (full length and oligomers) on mitochondrial matrix with MAM in physiological and pathological condition is important to dissect its role. For example, Figure S2A showed a mouse brain mitochondrial sub-compartmental fractions, however, there are no data showing the distribution of ATAD3 from AD mouse or DA1-treated mouse brain compared to WT mouse. Similarly, the distribution of ATAD3A on mitochondrial sub-compartmental fraction from A β treatment compared to control group is not clear.

Response: We thank the reviewer for the constructive comments. We have now provided data comparing the distribution of ATAD3A, obtained via mitochondrial fractionation, which included the MAM, under control and AD conditions. In the mitochondrial sub-compartmental fractions harvested from the cortex and hippocampus of WT and 5XFAD mice, the distribution of ATAD3A to the MAM fractions significantly elevated in 5XFAD mice compared with WT mice (Supplementary Fig. 3a in the revised manuscript). We also observed a significant increase in ATAD3A in the MAM fractions of Neuro2a cells treated with oligomeric A β ₁₋₄₂ (Supplementary Fig. 3a in the revised manuscript). Together, these results suggest that ATAD3A accumulates on the MAM in the context of AD, which is consistent with our PLA study in *in vitro* and *in vivo* models of AD.

We added the following sentences to the Results section in the revised manuscript (page 7).

“In the present study, mitochondrial sub-compartmental fractionation from mouse brains revealed ATAD3A enrichment in the MAM fractions; ATAD3A was present in the same mitochondrial fractions as VDAC and SigmaR1, two proteins that have been localized to the MAMs³⁵ (Supplementary Fig. 3a). Notably, the distribution of ATAD3A to the MAM fraction was significantly enhanced in 5XFAD mice compared with WT mice (Supplementary Fig. 3a). There was also a significant increase in ATAD3A on the MAM fractions of Neuro2a cells treated with oligomeric A β ₁₋₄₂ (Supplementary Fig. 3a).”

5. What is the sequence for TAT and DA1 peptide? It will be interesting to see the effect of the sequences from the first 50 amino acids as a control peptide to compare a truncated ATAD3A mutant.

Response: The sequences for TAT and DA1 peptide were published in our previous paper (Zhao *et al.*, *Nature Communications*, 2019). DA1 is the peptide homolog of the coiled-coil domain of ATAD3A, the domain required for oligomerization. Therefore, DA1 can serve as a decoy to block ATAD3A oligomerization.

ATAD3A is a nuclear-encoded mitochondrial protein that spans the inner and outer membranes; its two terminal domains are located in the outer membrane and the matrix. The ATAD3A mitochondrial transmembrane domain and C-terminal ATPase domain are required for ATAD3A mitochondrial localization (Gilquin *et al.*, *Mol Cell Biol.*, 2010). In our truncated ATAD3A variant studies, to maintain the mitochondrial distribution of ATAD3A, we generated a series of truncated mutants by deleting the N-terminal domain (e.g., the first 50 amino acids of the N-terminal domain or the coiled-coil domain) while preserving the transmembrane and c-terminal ATPase domains. In addition, in the present study, we aimed to assess the domain influence of mitochondria-localized ATAD3A on protein oligomerization and cellular phenotypes. Therefore, we do not use the first 50 amino acids for a comparison with the other variants.

6. It is clear the significant association of ATAD3 with CYP46A1 on APP process and cholesterol metabolism. However, the additional experiments are required to support that ATAD3 regulates amyloid pathology dependent on CYP46A1 in vivo, such as double transgenic ATAD3/CYP46A1 mice.

Response: The reviewer suggested examining the effects of CYP46A1 overexpression in ATAD3A heterozygous knockout AD mice. We thank the reviewer for this insightful comment. In 5XFAD mice, we showed that CYP46A1 was mainly decreased in neurons, which could be corrected by either DA1 treatment or ATAD3A heterozygous knockout. There are currently no commercially available CYP46A1-overexpressing mice. The generation of neuron-specific CYP46A1-overexpressing mice followed by their crossing with ATAD3A

heterozygous knockout 5XFAD mice (triple transgenic mice) would take us at least 2 to 3 years, if every step goes smoothly. Given the timeline for manuscript revision and the unavailability of the research material, we would greatly appreciate it if the reviewer allows us to not include these experiments in this manuscript. Instead, we have added the following text to the Discussion section in response to the reviewer's comment.

“Future studies of CYP46A1 overexpression in neurons of ATAD3A heterozygous knockout AD mice will further help to address the impact of neuronal CYP46A1 restoration on the MAM impairment, cholesterol metabolism dysregulation, and AD-associated neuropathology that are mediated by ATAD3A oligomerization.” (page 19 in the revised manuscript)

7. Rationale for using heterozygous or homozygous 5xFAD mice needs to be clarified.

Response: ATAD3A homozygous knockout mice are embryonically lethal, but heterozygous knockout mice are normal and fertile. We generated double-mutant 5XFAD^{het};CMV;ATAD3A^{fl/+} mice by crossing 5XFAD heterozygous mice with CMV;ATAD3A^{fl/+} mice. Thus, for the study using ATAD3A genetic AD mouse model, we need to compare 5XFAD heterozygous mice with 5XFAD^{het};CMV;ATAD3A^{fl/+} mice and wildtype littermates.

Homozygous 5XFAD mice represent a model with several advantages over heterozygous mice, developing the amyloid pathology much more rapidly, together with broader neurological phenotypes. Moreover, there are no gene-dosage effects in homozygous 5XFAD mice, which manifest in heterozygous 5XFAD mice. Thus, homozygous 5XFAD mice are a suitable model for examining the effects of pharmacological reagents on AD-associated neuropathology and behaviors. In response to the reviewer's comment, we have added the following text to the Results section of the revised manuscript.

“Compared with 5XFAD^{het}, 5XFAD^{homo} mice develop amyloid pathology much more rapidly, together with broader neurological phenotypes, and lack gene-dosage effects³⁹, making them more suitable for assessing the efficacy of DA1 treatment.” (page 11)

Reviewer #2 (Remarks to the Author):

In this manuscript, Zhao et al identify the oligomerization of ATAD3A, a mitochondria-associated AAA ATPase, as an Alzheimer's disease (AD) associated molecular anomaly that is induced by amyloid-beta and promotes the amyloidogenic pathway in vitro and in vivo. ATAD3A was shown to oligomerize and concentrate at the mitochondria-associated ER membranes (MAMs), where it represses the transcription of CYP46A1, an enzyme that mediates cholesterol hydroxylation and thus its catabolism. Consequently, cholesterol accumulates in AD models as a result of ATAD3A oligomerization and this accumulation promotes APP processing. Importantly, reducing ATAD3A oligomerization genetically in a haploinsufficient ATAD3A background or pharmacologically with a brain penetrant inhibitory peptide restores normal levels of CYP46A1 in CNS neurons from the 5XFAD mouse model, corrects cholesterol metabolism, and decreases amyloid burden, gliosis and cognitive deficits.

This is a thorough study on the pathogenic role of ATAD3A oligomerization in AD models that involves MAM dyshomeostasis, impact on cholesterol metabolism and APP processing. There are a lot of in vitro and in vivo data that support the hypothesis that CYP46A1 downregulation may account for these downstream phenotypes and a rather compelling case that downregulating ATAD3A genetically or preventing its oligomerization with a peptide confers benefits in the 5XFAD model. The missing links are: what mechanism causes ATAD3A oligomerization? What are the relevant “clients” of this AAA ATPase in the MAMs? How does ATAD3A oligomerization downregulate CYP46A1 transcription? Given the strength and large amount of data provided, these questions may be best suited for follow-up studies. However, there are a few aspects that need to be strengthened to make this manuscript suitable for publication in NCOMMs in my view.

1. The authors use the proximity ligation assay extensively to examine MAM proteins and results are overall rather compelling. The downside is that there is no information on ER or mitochondria morphology. Does

ATAD3A oligomerization induced by amyloid beta affect mitochondria fusion/fission? Can they show greater colocalization between ER and mitochondria with the appropriate ER and mitochondria markers?

Response: We previously reported that ATAD3A oligomerization causes excessive mitochondrial fission and induces mitochondrial fragmentation by recruiting the mitochondrial fission protein Drp1 to the mitochondria (Zhao *et al.*, *Nature Communications*, 2019). In response to the reviewer's comments, we examined the mitochondrial morphology of mouse hippocampal HT-22 neuronal cells with and without DA1 treatment. Addition of oligomeric A β induced mitochondrial fragmentation, which is consistent with previous studies. In contrast, DA1 treatment significantly reduced the percentage of cells with fragmented mitochondria (Fig. 1 in this response letter), supporting the notion that ATAD3A oligomerization induced by A β impairs mitochondrial dynamics and leads to mitochondrial fragmentation. Due to the limited space, we only present these data in the response letter.

To better visualize the colocalization between the ER and mitochondria in HT-22 cells, we co-stained the cells with antibodies against IP3R3 (an ER marker) and VDAC (a mitochondrial marker) antibodies and imaged the cells with super-resolution confocal microscope. We showed that A β treatment significantly enhanced the colocalization between IP3R3 and VDAC immunosignals (Supplementary Fig. 3b), consistent with our observation that toxic A β promotes ER and mitochondrial tethering (Fig. 2).

Fig. 1. DA1 treatment reduces A β -induced mitochondrial fragmentation. HT-22 cells were treated with DA1 or control TAT peptide (1 μ M), followed by the addition of oligomeric A β (5 μ M for 18 hours). The cells were then stained with anti-Tom20 antibody. The mitochondrial morphology of the cells was imaged and quantified. Data are mean \pm SEM of 3 independent experiments. At least 50 cells/group was counted.

2. Does ATAD3A oligomerization increase its AAA ATPase activity? In other words, is it a gain of function that results in its pathogenic effect? What is the impact of an ATPase-dead mutant on MAMs?

Response: In response to the reviewer's comments, we performed the following experiments and concluded that ATAD3A oligomerization does not affect its AAA-ATPase activity and vice versa. Specifically,

- 1) We immunoprecipitated ATAD3A from six-month-old WT and 5XFAD mouse cortex with anti-ATAD3A antibodies, and then assessed ATPase activity in the immunoprecipitants. We found that ATAD3A ATPase activity was comparable in six-month-old WT and 5XFAD mouse brains. The immunoprecipitation analysis also demonstrated no interaction between ATAD3A and other ATPase proteins such as VCP and LonP1, ruling out the influence of other mitochondrial AAA ATPases (Supplementary Fig. 2d in the revised manuscript).
- 2) We immunoprecipitated Flag from Neuro2a cells expressing ATAD3A-WT-Flag, ATAD3A- Δ N50-Flag or ATAD3A- Δ CC-Flag. After ATPase activity measurement in the immunoprecipitants, we found no change in ATAD3A ATPase activity among the experimental groups (Supplementary Fig. 2f in the revised manuscript).
- 3) We overexpressed ATAD3A-WT-Flag (ATAD3A wildtype) or ATAD3A-K358E-Flag (an ATPase dead mutant) plasmids and then assessed ATAD3A oligomerization. Overexpression of ATAD3A-WT-Flag enhanced ATAD3A oligomerization, whereas ATAD3A-K358E-Flag overexpression did not affect ATAD3A oligomerization (Supplementary Fig. 2e in the revised manuscript).

The above-described data are provided in Supplementary Fig. 2. The following sentences were added to the Results section.

“There was no change in ATAD3A ATPase activity in the cortex of 5XFAD mice relative to wildtype (WT) mice (Supplementary Fig. 2d). Overexpression of ATAD3A ATPase dead mutant ATAD3A-K358E-Flag³² also had no

effects on ATAD3A oligomerization (Supplementary Fig. 2e). Thus, ATAD3A oligomer formation is not associated with enzyme activity of the protein.” (page 6)

“The expression of ATAD3A-WT-Flag or ATAD3A-ΔN50-Flag did not alter the levels of the MAM-related proteins (Supplementary Fig. 3e), nor ATAD3A ATPase activity (Supplementary Fig. 2f).” (page 8)

To determine the effects of ATAD3A ATPase dead mutant on MAM integrity, we transduced HT-22 cells with ATAD3A-WT-Flag or ATAD3A-K358E-Flag and examined MAM tethering. Overexpression of ATAD3A-WT-Flag significantly enhanced the number of PLA-positive puncta when HT-22 cells were stained with anti-SigmaR1 and anti-VDAC antibodies or anti-ATAD3A and anti-FACL4 antibodies. However, overexpression of ATAD3A-K358E had no effects on MAM integrity. The following sentences were added to the revised manuscript.

“In contrast, expression of ATAD3A-K358E-Flag, an ATAD3A ATPase dead mutant, did not affect MAM tethering (Supplementary Fig. 3f), further supporting the notion that the biological effects of ATAD3A oligomers do not result from its enzymatic activity.” (page 8)

3. The DA1 peptide was used before in another in vivo study. Can the authors provide drug levels in the CNS of the 5XFAD mice? It’s always difficult to interpret efficacy data without knowing anything about drug levels.

Response: Thank you for the reviewer’s comment. To analyze the PK/PD of DA1 peptide in mice, we would have to subcontract the work out to a contract research organization. The cost would be a minimum of \$100,000. Due to limited funds, we cannot afford such an expensive experiment. Meanwhile, working with the Mass Spectrometry

Core Facility at Cleveland Clinic Foundation Research Center, we attempted to determine the presence of DA1 after DA1 treatment in mice. We found that, 10 minutes following the treatment (intravenous injection of 10 mg/kg DA1), the peptide DA1 could be detected in samples of total brain lysates and plasma (Fig. 2a-d in this response letter). In addition, in mice treated with FITC-conjugated DA1 peptide, we were able to detect the FITC fluorescent signal in the brain cells 1 day after the peptide treatment, further indicating the brain permeability of the DA1 (Fig. 2e in this response letter). We only present these data in the response letter to respond to the reviewer’s comments. Currently, we are seeking funding to support the PK/PD analysis of DA1 in mice, in order to thoroughly understand the pharmacological characteristics of DA1 in AD mice.

Fig. 2. DA1 enters the brains of AD mice. LC/MS/MS Chromatograms of DA1 in mouse Samples. (a) DA1 at six charged in positive mode was analyzed on a triple quadrupole mass spectrometer (TSQ Quantiva, Thermo Fisher Scientific) using Selective Reaction Monitoring (SRM) with the transition at m/z 406.2 > 257.5. The LC/MS/MS chromatograms of DA1 in samples are shown for (b) DA1 added to the brain homogenate, (c) DA1 detected in plasma and (d) DA1 detected in brain homogenate after intravenous administration of DA1 (10 mg/kg) in mice. (e) Wildtype mice were treated with FITC-conjugated DA1 using an osmotic minipump. One day after treatment, mouse brains were harvested. Mouse brain sections were imaged by microscopy. FITC-conjugated DA1 (green) was observed in the brain cells one day after treatment.

4. Does Efavirenz, a CYP46A1 agonist, correct the phenotypes induced by ATAD3A oligomerization *in vitro*?

Response: Per the reviewer's suggestion, we conducted several experiments to test the effects of efavirenz on cellular phenotypes induced by ATAD3A oligomerization *in vitro*. We can summarize the findings as follows.

- 1) Efavirenz treatment significantly reduced the cholesterol content that was induced by expression of ATAD3A variants (Supplementary Fig. 6k in the revised manuscript).
- 2) In stable APP wildtype (APP^{wt})-expressing Neuro2a cells, overexpression of ATAD3A-WT or -ΔN50 elicited APP cleavage, which was significantly reduced upon efavirenz treatment (Supplementary Fig. 7g in the revised manuscript).
- 3) Overexpression of ATAD3A-WT or -ΔN50 variant caused the loss of synaptic density in primary neurons. Treatment with efavirenz significantly improved the synaptic integrity (Supplementary Fig. 8c in the revised manuscript).

Taken together, these results indicate CYP46A1 as a downstream effector of ATAD3A oligomerization and demonstrate the protective role of efavirenz in correcting phenotypes induced by ATAD3A oligomers.

The following sentences were added to the Results section in the revised manuscript.

“Similarly, treatment with efavirenz (EFV), an allosterical activator of CYP46A1, also abolished ATAD3A-WT- and -ΔN50-induced cholesterol accumulation (Supplementary Fig. 6k).” (page 15)

“These results are consistent with our observation that EFV treatment abolished the ATAD3A-WT- or -ΔN50-induced increase in C99 production in stable APP^{wt} Neuro2a cells (Supplementary Fig. 7g).” (page 16)

“Similarly, enhancement of CYP46A1 by EFV treatment abolished ATAD3A-WT or ATAD3A-ΔN50-induced synaptic loss in primary neurons (Supplementary Fig. 8c).” (page 17)

Minor comments

5. Page 3, 2nd paragraph: *“Moreover, the amyloid precursor protein (APP) and APP processing γ -secretases, presenilin-1 (PS1) and presenilin-2 (PS2), are highly enriched in the MAMs relative to other cell compartments, such as the plasma membrane, mitochondria, and ER”. I don't think that APP is enriched in MAMs compared to other compartments. It is also abundant in the TGN, plasma membrane and endosomal compartment so perhaps a more conservative view is that APP is “present in the MAMs”. Presenilins, on the other hand, may well be enriched in the MAMs.*

Response: We have corrected the sentence as below.

“Moreover, the amyloid precursor protein (APP) processing γ -secretases, presenilin-1 (PS1) and presenilin-2 (PS2), are highly enriched in the MAMs relative to other cell compartments, such as the plasma membrane, mitochondria, and ER.” (page 3)

6. Can the authors provide western blot data showing that the 5XFAD brains have lower CYP46A1 protein levels, consistent with the proteomics?

Response: We provide western blot data in the revised manuscript (Supplementary Fig. 6b) and have added the following sentence to the Results section.

“The decreased protein level of CYP46A1 in 5XFAD^{het} mouse brains was also restored by heterozygous knockout of ATAD3A (Supplementary Fig. 6b).” (page 13)

Reviewer #3 (Remarks to the Author):

In this study, the authors showed ATAD3A oligomerization by amyloid beta is critical in AD pathogenesis by regulating cholesterol turnover and MAM integrity, thereby exacerbating AD pathology. Although genetic ATAD3A deficiency and pharmacological inhibition to inhibit ATAD3A oligomers reduced AD pathology in AD model mice, there were some problems that need to be addressed.

Major points:

1. The authors discovered that the presence of amyloid beta (in vitro, AD patients, and 5XFAD mice) induces the oligomerization of ATAD3A and the levels of ATAD3A monomer does not change when using beta-mercaptoethanol to limit the oligomerization. However, the immunostaining signals of ATAD3A in AD patients and AD model mice are markedly different compared to controls. If so, is this antibody recognizing oligomers of ATAD3A? If the antibody against ATAD3A used by the authors could detect the entire ATAD3A form, including monomers and oligomers, then the staining pattern of ATAD3A would be expected to be similar in both control and AD conditions. Therefore, it seems likely that the antibody will need to be validated using other antibodies whether it correctly recognizes ATAD3A.

In addition, the staining pattern of ATAD3A is distributed throughout the cytoplasm, not in the form of mitochondria or MAMs. And in the AD context, it appears that ATAD3A expression was induced rather than the oligomerization of ATAD3A.

Response: In our immunostaining studies, we used the ATAD3A antibody from Abcam (ab112572), which targets the N-terminus of ATAD3A. We validated the specificity of the antibody by knocking down ATAD3A in cells (see Fig. 3a in this response letter), and in 5XFAD mouse brain (see Supplementary Fig. 4a in the revised manuscript). We also co-stained endogenous ATAD3A protein with the antibody and mitochondria with MitoTracker Red in HT-22 cells. The pattern of endogenous ATAD3A recognized by the ATAD3A antibody exclusively colocalized with the mitochondrial network (Fig. 3b in the response letter). Thus, the antibody for ATAD3A staining functions well and has been used in our previous studies (Zhao *et al.*, *Nature Communications* 2019). In parallel, we stained the postmortem brain sections of AD patients and normal individuals with another ATAD3A antibody (GeneTex, GTX116301). Consistent with our results shown in the manuscript (Fig. 1f), the immunodensity of ATAD3A recognized by the Genetex antibody was higher in AD patient brains than in normal subjects (See Fig. 3c in the response letter). The staining antibody used here recognizes both monomeric and oligomeric forms of ATAD3A by Western blotting. Unlike Western blotting, immunostaining of ATAD3A in brain tissue and cell cultures shows *in situ* signals that identify the epitope of the *in vivo* state of ATAD3A (here, based on our extensive data, it can be considered the oligomeric state of ATAD3A). This is similar to the cases of some amyloid-related antibodies that recognize both amyloid precursor protein and oligomeric amyloid β by Western blotting, whereas in immunostaining of brain tissues they show stronger immunosignals for oligomeric A β .

Our original manuscript provided extensive results revealing that ATAD3A oligomers were increased in various AD experimental models and that treatment with DA1, a selective inhibitor of ATAD3A oligomerization, significantly rescued AD-related phenotypes (Figs. 4-7). In contrast, the total mRNA and protein levels of ATAD3A were comparable between wildtype and 5XFAD mouse brains (Supplementary Fig. 2 in the revised

Fig. 3. ATAD3A immunostaining. a) MEF cells were transfected with control or ATAD3A shRNA. The cells were stained with anti-ATAD3A antibody (Abcam, ab112572). b) HT-22 cells were stained with ATAD3A antibody (Abcam) and MitoTracker Red. c) Postmortem brain sections of AD patients and normal individuals were stained with anti-ATAD3A antibody (GeneTex, GTX116301).

manuscript), or AD cell culture models (Fig. 1a and Supplementary Fig. 2a in the revised manuscript). Our response to Major Concern #1 of Reviewer #1 can also be referred to in order to support our contention that ATAD3A oligomerization rather than expression was induced to affect MAM integrity and cholesterol accumulation in AD. To respond to the reviewer's comment, we further show that ATAD3A immunodensity in 5XFAD mice was reduced by DA1 treatment (Fig. 4a in the response letter). In HT-22 cells exposed to toxic A β , the increasing immunodensity of ATAD3A was abrogated by DA1 treatment (Fig. 4b in the response letter). Again, the total protein levels of ATAD3A were not altered in all experimental groups under the same conditions (Fig. 4c in the response letter). Collectively, these lines of evidence suggest that the elevated immunodensity of ATAD3A in the AD patient and mouse brains is most likely due to increased ATAD3A oligomerization.

Due to limited space, we only present these data in the response letter. In the original manuscript, we have provided a statement to discuss this immunodensity changes in ATAD3A in AD patients and AD mice as follows.

“Thus, the elevated immunodensity of ATAD3A in AD patient and mouse brains is likely due to increased ATAD3A oligomerization, consistent with our previous observation²⁷.” (page 6 in the revised manuscript)

Fig. 4. DA1 treatment reduces ATAD3A immunodensity in AD mice and in cells. (a) wildtype and 5XFAD mice were treated with DA1 or control peptide TAT starting from 6 weeks. Mouse brains were harvested at the age of 6 months and brain sections were stained with anti-ATAD3A antibody (Abcam ab112572). The immunodensity of ATAD3A was quantified. Scale bar: 20 μ m. n=5 mice/group. (b) HT-22 cells were treated with DA1 or TAT (1 μ M) followed by treatment with oligomeric A β (5 μ M, 12h). Cells were then stained with anti-ATAD3A antibodies, and the immunodensity of ATAD3A was quantified. Scale bar is 20 μ m. Data were from 3 independent experiments. (c) Total protein lysates of the indicated groups were subjected to Western blotting with anti-ATAD3A antibody. Actin was used as a loading control. Data were from 3 independent experiments. All data are expressed as mean \pm SEM.

2. In Figure 2E, the integrity of MAM appears to be affected only by the knockdown of ATAD3A regardless of the presence of amyloid beta. It appears that the function of ATAD3A itself, not oligomerization, has a significant impact on MAM integrity. There appears to be insufficient evidence for the effect of ATAD3A oligomerization on MAM integrity.

Response: In our original manuscript, we provided data showing that treatment with DA1, a selective inhibitor of ATAD3A oligomerization, significantly reduced the number and size of VDAC/IP3R3 PLA puncta and VDAC/SigmaR1 PLA puncta in mouse hippocampal neuronal HT-22 cells exposed to toxic A β (Supplementary Fig. 4b in the original manuscript). These results support our conclusion that MAM integrity was mediated by A β -induced

ATAD3A oligomerization. In the revised manuscript, we have moved the data to Supplementary Fig. 3d, and modified the following sentences (page 8).

“We knocked down ATAD3A in HT-22 cells using lentiviral ATAD3A shRNAs or treated the cells with DA1 peptide that we developed to block ATAD3A oligomerization²⁷. In the presence of oligomeric A β ₁₋₄₂ peptides, either ATAD3A downregulation or DA1 treatment significantly reduced the number of PLA-positive puncta in A β -treated HT-22 cells stained with anti-IP3R3 and anti-VDAC antibodies compared to control groups (Fig. 2d, Supplementary Fig. 3d).”

3. The authors used a mutant form of ATAD3A (delta N50) to promote the oligomerization of ATAD3A. In results where authors used the mutant, the effect of N50 does not show a significant effect compared to WT (no statistical significance), but rather it seems to be an overexpression effect of ATAD3A. Again, it is difficult to say that this is a phenomenon caused by oligomerization.

Response: In our study, overexpression of ATAD3A- Δ N50-Flag significantly promoted cellular phenotypes when compared with those of cells expressing ATAD3A-WT-Flag. In the revised manuscript, we provided the p values in Figs. 2f, 5j, 5l, 5m, 6c, 7a and Supplementary Figs. 6j, 6k, 7g and 8c to show the statistical significance between WT and Δ N50 group. In addition, we transfected the cells with ATAD3A- Δ CC-Flag, which abolishes ATAD3A oligomer formation, and found that overexpression of ATAD3A- Δ CC-Flag corrected the cellular phenotypes (cholesterol accumulation and APP processing) to the levels similar to those in control vector-expressing cells (see Fig. 6c and Supplementary Fig. 6j).

4. Representative staining images are missing in Figures 3E, 4B, and 5H. If the space for results is insufficient, it also should be added in the supplementary figure.

Response: We have now provided representative images in the revised manuscript (Supplementary Figs. 4e, 5c and 6c).

5. It seems necessary to confirm the staining pattern of Jade C in Figure 4G. As Jade C-positive signals are known to be mainly present on the cell body, it is difficult to find a co-localized pattern confined to DAPI in the result. Authors need to determine whether the staining results are indicative of actual degenerating neurons.

Response: To examine the neuronal nature of degenerative cells, we co-stained brain sections of TAT- and DA1-treated 5XFAD mice with FJC dye and anti-NeuN (a neuronal marker) antibody, a method that has been used in other studies [Wang et al., *Time-course of neuronal death in the mouse pilocarpine model of chronic epilepsy using Fluoro-Jade C staining. Brain Res 1241,157-167(2008)*]. We observed a large number of NeuN⁺ neuronal cells in the hippocampus and cortex of 5XFAD mice that were immunopositive for FJC staining. In contrast, the density of FJC fluorescence signals in NeuN⁺ cells in DA1-treated 5XFAD mice was greatly reduced. The revised figures are presented in Fig. 4g in the revised manuscript. The following sentences were added to the revised manuscript.

“We stained brain sections of six-month-old 5XFAD^{homo} mice with the Fluoro-Jade C (FJC) fluorescent probe that selectively binds to degenerating neurons⁴² and anti-NeuN antibody. We observed FJC-positive fluorescence signals in NeuN⁺ cells in the CA1 region of the hippocampus and cortex, indicating ongoing neuronal loss (Fig. 4g).” (page 11)

6. Figure 5B shows staining of CYP46A1 in the hippocampal CA1 region. In this regard, Figure 1 did not show the staining results of ATAD3A in the hippocampal CA1, but it would be nice if it showed pattern changes of ATAD3A in the hippocampus.

Response: We have provided data showing the change in ATAD3A in the hippocampus of 5XFAD mice (Supplementary Fig. 2g in the revised manuscript).

Minor points:

1. The authors used the phrase 'ATAD3A activation' in the middle of the paper, so the concepts of 'oligomerization' and 'activation' seem confusing. It should be unified through 'oligomerization'.

Response: We have corrected the word in question and used “oligomerization” throughout the revised manuscript.

2. The authors need to modify the color index of the heatmap in Figure 5A. In general, red indicates an increase and blue indicates a decrease. Also, the authors need to write the value of the color index.

Response: The heatmap was revised accordingly and is now shown in Fig. 5a and Supplementary Fig. 6a.

Reviewers' Comments:

Reviewer #1:

Remarks to the Author:

Authors well reponds to my comment in the revision.

Reviewer #2:

Remarks to the Author:

The authors have addressed all my concerns, except for an important one related to Figure 4: I think it's important to show the drug DA1 gets into the brain and while I understand that proper PK/PD experiments may be very expensive, the authors can quantify the FITC fluorescence they show in Figure 2e from the response letter. Uninjected control brains should also be quantified.

Reviewer #3:

Remarks to the Author:

The authors answered all comments through sufficient explanations and experiments.

Response letter

RE: ATAD3A oligomerization promotes neuropathology and cognitive deficits in Alzheimer's disease models (manuscript reference NCOMMS-21-14370A)

We thank the reviewers for their positive comments on our revision and constructive suggestions. In the following sections, we provide point-by-point responses to these comments. We believe that these changes have significantly improved our manuscript and we hope that the editor and reviewers will now find the revised manuscript suitable for publication.

REVIEWER COMMENTS

Reviewer #1 (Remarks to the Author):

Authors well repond to my comment in the revision.

Response: We thank reviewer #1 for the positive comments on our revisions.

Reviewer #2 (Remarks to the Author):

The authors have addressed all my concerns, except for an important one related to Figure 4: I think it's important to show the drug DA1 gets into the brain and while I understand that proper PK/PD experiments may be very expensive, the authors can quantify the FITC fluorescence they show in Figure 2e from the response letter. Uninjected control brains should also be quantified.

Response. We thank the reviewer for the comments and understanding of the current difficulties with PK/PD experiments. As suggested by the reviewer, we provided quantitative data on the fluorescence density of FITC-conjugated DA1 in the brains of treated mice. Uninjected control mice were used as controls. Our result showed that FITC-conjugated DA1 fluorescence signals significantly accumulated in the wildtype mouse brains one day after osmotic pump implantation (See figure on the right).

We previously reported that DA1 administered subcutaneously can enter the brains of two transgenic mouse models and reduce neuropathologies and behavioral deficits associated with Huntington's disease (*Zhao et al., Nature Communications 2019*). In the current study, we demonstrate that subcutaneous treatment with DA1 is neuroprotective in a mouse model of Alzheimer's disease by blocking ATAD3A oligomerization. These findings in multiple *in vivo* animal models confirm that DA1 can enter the brain and act as a specific inhibitor of ATAD3A oligomerization to reduce neurodegeneration. Thus, DA1 or DA1-like reagents might be a useful therapeutic strategy for preventing or treating neurodegenerative diseases, such as AD.

We now provided the data in the Supplementary Fig. 5b (top panel). The figure legend is provided in the Supplementary information (Supplementary Fig. 5b). The following sentences are added to the revised manuscript (page 11).

“In addition, FITC-conjugated DA1 fluorescence signals significantly accumulated in the brains of WT mice one day after osmotic pump implantation (Supplementary Fig. 5b), confirming that DA1 administered subcutaneously can enter mouse brains.”

Reviewer #3 (Remarks to the Author):

The authors answered all comments through sufficient explanations and experiments.

Response: We thank reviewer #3 for the positive comments on our revisions.

Reviewers' Comments:

Reviewer #2:

None